# AGENTS AS KNOWLEDGE INTEGRATOR AND UTILIZER IN MULTIMODAL RECOMMENDATION

## ABSTRACT

The proliferation of online multimodal content has driven the adoption of multimodal data in recommendation systems. Current studies either enhance item features with multimodal data or construct additional homogenous graphs via multimodal data. However, a significant semantic gap exists between multimodal data and recommendation tasks. This gap introduces modality-specific noise irrelevant to recommendation tasks when enhancing item features and results in homogenous graphs built on multimodal data that fail to adequately consider users' historical behaviors. Fortunately, the multimodal information understanding and contextual processing capabilities of large language models (LLMs) have emerged as a promising approach to bridging this semantic gap.

To this end, we propose AgentMMRec, a novel agent-based framework that bridges the semantic gap via two cooperative agents: an Integrator Agent that uses LLMs to infer user preferences and item properties from multimodal data and users' historical behaviors, storing knowledge in a knowledge memory; and a Utilizer Agent that refines traditional homogenous item-item graphs using there knowledge, constructs behavior- and multimodal-aware homogenous graphs, and performs knowledge-enhanced reranking in recommendation stage. Integrator Agent updates the memory based on feedback from reranking performance. Extensive experiments on real-world datasets demonstrate that AgentMMRec outperforms existing multimodal recommendation models and exhibits superior performance across various data sparsity scenarios. Additionally, AgentMMRec can enhance the performance of existing multimodal recommendation models by leveraging the constructed knowledge memory. Code can be found in anonymous link[1].

## 1 INTRODUCTION

The exponential growth in the variety and volume of online information has made leveraging multimodal data to enhance recommender systems a mainstream paradigm (Xu et al., 2025d; Chen et al., 2025b; Zhou et al., 2023a). Current multimodal recommendation studies (Xu et al., 2025c; Zhou & Shen, 2023; Xu et al., 2025a; Zhang et al., 2022) primarily focus on two approaches: enhancing the explicit features of items using multimodal data and constructing additional homogeneous graph structures based on multimodal data to improve performance. However, a significant semantic gap exists between multimodal data and recommendation tasks (Xu et al., 2025f; Liu et al., 2024). This gap introduces modality-specific noise irrelevant to the recommendation task when enhancing item features and results in homogeneous graphs built on multimodal data that fail to adequately consider users' historical behaviors.

Recently, many studies (Wei et al., 2024; Ren et al., 2024; Fioretti et al., 2025; Xu et al., 2025b; Bao et al., 2023) in multimodal recommendation have attempted to leverage the multimodal understanding and contextual processing capabilities of large language models (LLMs). Current mainstream studies can be broadly categorized into three paradigms: a) LLM for data augmentation (Xu et al., 2025b; Wei et al., 2024), b) LLM as a backbone model with fine-tuning (Bao et al., 2023; Zhang et al., 2025a), and c) LLM as a reranker (Hou et al., 2023; 2024). However, these paradigms have notable limitations. Paradigm (a) leaves the multimodal data constrained to its inherent properties, lacking sufficient alignment with the recommendation task. Paradigm (b) is both cost-prohibitive and limited by the

---

[1] https://anonymous.4open.science/r/AgentMMRec-r/

small amount of task-specific data available in recommendation scenarios, making it difficult for LLMs to effectively fit the task. Paradigm (c), while reranking items, solely depends on partial item information in final list and still fails to fully leverage users' historical behaviors. The multimodal understanding capabilities of LLMs enable them to fully comprehend and utilize multimodal data, while their contextual processing capabilities provide significant advantages in processing users' historical behaviors. Therefore, a comprehensive agents paradigm that fully leverages the multimodal understanding and contextual processing capabilities of LLMs to bridge the semantic gap between multimodal data and recommendation tasks has become a promising and urgently needed solution.

To this end, we propose a novel agent-based framework (AgentMMRec), which consists of two tailored agents with distinct roles. Specifically, Integrator Agent leverages the multimodal understanding capabilities of LLMs as a knowledge integrator, while Utilizer Agent employs the contextual processing abilities of LLMs as a knowledge utilizer. Together, these agents bridge the semantic gap between multimodal data and recommendation tasks, thereby enhancing the performance of multimodal recommendations. More specifically, Integrator Agent uses multimodal data about items and the contextual reasoning capabilities of LLMs to infer user preferences and item properties based on users' historical behaviors and item information. These inferences are stored in the knowledge memory. Furthermore, Utilizer Agent refines traditional homogeneous item-item graphs using the knowledge in the knowledge memory. It also employs the multimodal understanding capabilities of LLMs and the knowledge from the constructed knowledge memory to build additional behavior- and multimodal-aware homogenous that fully consider users' historical behaviors and multimodal data. During the recommendation phase, Utilizer Agent reranks the final recommendation list based on the knowledge stored in the knowledge memory and provides performance feedback to Integrator Agent, which updates the constructed knowledge memory based on this feedback.

Extensive experiments on multiple real-world datasets demonstrate that AgentMMRec outperforms existing multimodal recommendation models and exhibits superior performance across various data sparsity scenarios. Additionally, the agents in AgentMMRec can enhance the performance of existing multimodal recommendation models by leveraging the constructed knowledge memory. It is worth noting that, due to the presence of the constructed knowledge memory, knowledge memory continuously improves through multiple rounds of updating with AgentMMRec or relay updating with multiple different models and demonstrates significant effectiveness in handling cold-start items. The main contributions of this work can be summarized as follows:

- We identify the semantic gap between multimodal data and recommendation in multimodal recommendations and further point out the limitations for exisiting LLM-based solutions.

- We propose AgentMMRec, a novel agent-based multimodal framework, which design two specialized agents to leverage the multimodal understanding and contextual processing capabilities of LLMs to bridge the semantic gap between multimodal data and recommendation tasks.

- We conducted extensive experiments to validate the effectiveness of our AgentMMRec. Moreover, we validated the integration capability of AgentMMRec with existing models, as well as its effectiveness in scenarios with varying data sparsity, cold-start setting.

## 2 RELATED WORK

### 2.1 MULTIMODAL RECOMMENDATION

Recent researches have integrated multimodal data to address data sparsity in recommendation systems. A notable milestone was achieved by VBPR (He & McAuley, 2016), which incorporated visual content into matrix factorization (Rendle et al., 2009), using item images to enhance recommendations. Building on this, subsequent studies (Chen et al., 2019; Liu et al., 2019; Yu et al., 2023; Chen et al., 2025a) combined visual and textual modalities to enrich item representations and improve system effectiveness. More recently, MMGCN (Wei et al., 2019) pioneered the use of Graph Convolutional Networks (GCNs) to extract modality-specific features from user-item interactions. Models like DualGNN (Wang et al., 2021) and LATTICE (Zhang et al., 2021) introduced user-user and item-item graphs to capture shared preferences and relationships. Building upon LATTICE, FREEDOM (Zhou & Shen, 2023) improved representation stability by freezing item semantic graphs and reducing noise in user-item bipartite graphs. More recently, self-supervised learning and inter-modal relationships have gained traction. MMSSL (Wei et al., 2023) and MENTOR (Xu et al., 2025e) used contrastive

self-supervised learning to align multimodal inputs with collaborative signals, achieving strong results without requiring extensive labeled data. BM3 (Zhou et al., 2023b) explored inter-modal relationships to improve both recommendation accuracy and modality fusion. Additionally, LGMRec (Guo et al., 2024) utilized hyper-graph structures to model complex global and local relationships, while COHESION (Xu et al., 2025c) introduced a dual-stage fusion mechanism to enhance multimodal recommendation performance.

Despite these advancements, recent surveys (Xu et al., 2025f; Liu et al., 2024) highlight that a significant challenge in multimodal recommendation systems is the semantic gap between multimodal data and recommendation tasks. While some studies (Xu et al., 2025e; Zhou et al., 2023b; Wei et al., 2023) have attempted to align features across modalities, the lack of contextual understanding and comprehensive multimodal processing has limited the effectiveness of rigid alignment methods, leaving room for further improvement.

### 2.2 LLM-based Recommendation

Recently, LLMs have gained significant attention for their exceptional multimodal understanding and contextual processing capabilities. Numerous studies (Wei et al., 2024; Tian et al., 2023; Bao et al., 2023; Hou et al., 2023; Lee et al., 2024; Zhang et al., 2025b; Wei et al., 2024) have explored leveraging LLMs to enhance recommendation performance. For instance, TALLRec (Bao et al., 2023) adopts an instruction fine-tuning framework using the LLaMA model (Touvron et al., 2023). LEARN (Zhang et al., 2025b) integrates key attributes like title, description, and brand into predefined prompts and utilizes the LLM's last-layer features as item embeddings. LLMRank (Hou et al., 2023) formulates recommendation as a conditional ranking task, where sequential interaction history serves as the condition and retrieved items as candidates, which are then reranked by the LLM. Similarly, LLMRec (Wei et al., 2024) addresses sparse feedback and low-quality side information by analyzing user preferences and item attributes. Other works, such as (?Zhang et al., 2025a), attempt to fine-tune LLMs for recommendation tasks to optimize performance.

However, most LLM-based recommender systems primarily focus on directly utilizing LLMs' multimodal understanding and contextual processing or treating them as backbones for recommendation models. While these approaches show promise, they fail to address the broader challenges of multimodal recommender systems. Specifically, these studies overlook the potential of LLMs to bridge the semantic gap between multimodal data and recommendation tasks through deeper multimodal understanding and contextual reasoning.

To this end, our AgentMMRec leverages the seamless collaboration of two agents to liberate the multimodal understanding and contextual processing capabilities of LLMs, thereby bridging the gap between multimodal data and recommendation tasks.

## 3 Methodology

As illustrated in Figure 1 and Algorithm 1 in Appendix A.2, AgentMMRec consists of two tailored agents—Integrator Agent (IAgent($\cdot$)) and Utilizer Agent (UAgent($\cdot$))—along with a knowledge memory for knowledge retention. Integrator Agent constructs novel behavior- and multimodal-aware homogeneous graphs based on users' historical behaviors and multimodal data. It then extracts knowledge related to user preferences and item properties, storing these knowledge in the knowledge memory. Utilizer Agent then leverages the constructed knowledge memory to refine the traditional homogeneous item-item graphs built from multimodal data. During the recommendation stage, Utilizer Agent integrates the knowledge memory to re-rank the final recommendation list and provides performance feedback to the Integrator Agent. This feedback enables Integrator Agent to update the knowledge memory, ensuring continuous improvement in knowledge quality.

### 3.1 Problem Definition

Formally, let $\mathcal{U} = \{u_1, \ldots, u_{|\mathcal{U}|}\}$ and $\mathcal{I} = \{i_1, \ldots, i_{|\mathcal{I}|}\}$ be the set of users and items, respectively. Each item $i$ includes textual data (Title: $T_i^{title}$, Brand: $T_i^{brand}$, Categories: $T_i^{categories}$, and Description: $T_i^{description}$) and visual data (Image: $V_i$). Most advanced existing multimodal recommendation models (Chen et al., 2025a; Zhou & Shen, 2023; Guo et al., 2024; Xu et al., 2025f) directly utilizing

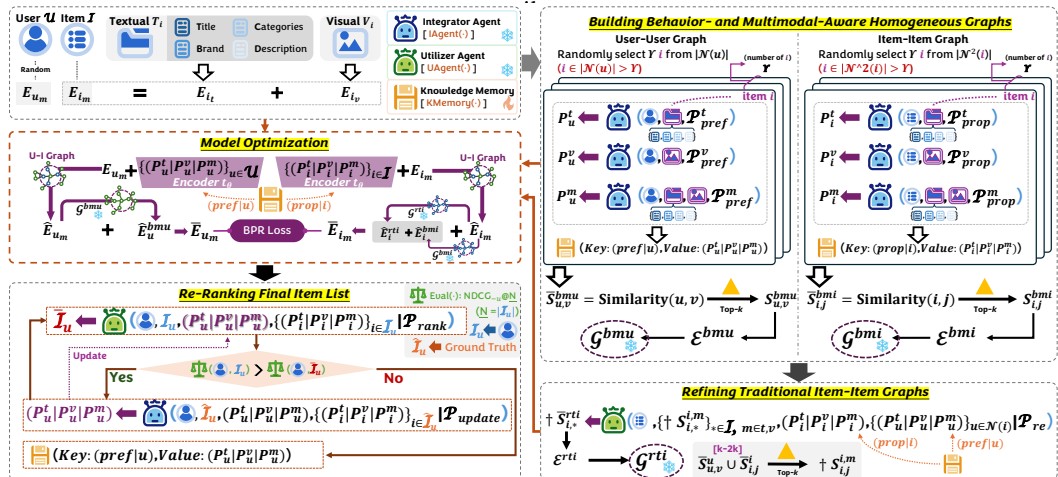

Figure 1: Overview of AgentMMRec. All modules correspond strictly to methodology.

MMRec[2] to encode textual and visual data using a pretrained sentence transformer, denoted as $t_\theta(\cdot)$, and a convolutional neural network (CNN), denoted as $v_\theta(\cdot)$. Formally, the textual representation of item $i$, $\mathbf{e}_{i_t}$, is computed as: $\mathbf{e}_{i_t} = t_\theta(T_i)$, where $T_i = (T_i^{\text{title}}|T_i^{\text{brand}}|T_i^{\text{categories}}|T_i^{\text{description}})$ represents the concatenation (denoted by $|$) of the item's title, brand, categories, and description. Similarly, the visual representation of item $i$, $\mathbf{e}_{i_v}$, is computed as: $\mathbf{e}_{i_v} = v_\theta(V_i)$. The entire item representations for modality $m \in t, v$ can be denoted as $\mathbf{E}_{i_m} \in \mathbb{R}^{d_m \times |\mathcal{I}|}$, where $d_m$ represents hidden dimensionality of modality $m$. Entire user representations for each modality $m$ are randomly initialized as $\mathbf{E}_{u_m} \in \mathbb{R}^{d_m \times |\mathcal{U}|}$. To ensure the fairness of comparisons, we also utilize the encoders provided by MMRec in our AgentMMRec. The user-item interaction matrix is denoted as $\mathcal{R} \in \{0, 1\}^{|\mathcal{U}| \times |I|}$. Specifically, each entry $\mathcal{R}_{u,i}$ indicates whether the user $u$ is connected to item $i$, with a value of 1 representing a connection and 0 otherwise. This matrix naturally constructs the bipartite graph $\mathcal{G} = (\mathcal{U}, \mathcal{I}, \mathcal{E})$, where $\mathcal{U}, \mathcal{I}$ serve as vertices, and $\mathcal{E}$ denotes the edge set. For each user-item pair $(u, i)$ that satisfies $\mathcal{R}_{u,i} = 1$, there exists bidirectional edges $(u, i) \in \mathcal{E}$ and $(i, u) \in \mathcal{E}$. Notably, unlike multimodal sequential recommendation, multimodal recommendation does not have access to the temporal order and dynamic evolution of user behaviors. As a result, multimodal recommendation scenarios place greater emphasis on accurately capturing user preferences and item properties.

## 3.2 BUILDING BEHAVIOR- AND MULTIMODAL-AWARE HOMOGENEOUS GRAPHS

Integrator Agent extracts users' preferences and items' properties by combining user-item historical interactions with multimodal data and storing these knowledge in the knowledge memory. Based on these knowledge, Integrator Agent then constructs a behavior- and multimodal-aware user-user Graph and a behavior- and multimodal-aware item-item Graph.

### 3.2.1 BEHAVIOR- AND MULTIMODAL-AWARE USER-USER GRAPH

Since users typically lack multimodal data, previous multimodal recommendation models (Zhou et al., 2023a; Xu et al., 2025f) mostly do not construct a user-user homogeneous graph or only build user-user graphs based on historical interactions (Wang et al., 2021; Xu et al., 2025c), failing to leverage multimodal data effectively. Benefiting from the multimodal understanding and contextual processing capabilities of LLMs, the Integrator Agent generates behavior- and multimodal-aware preferences $P_u$ for each user $u \in \mathcal{U}$. This is achieved by feeding the multimodal data of items interacted with by user $u$, along with carefully designed prompt templates $\mathcal{P}_{pref}^t$, $\mathcal{P}_{pref}^v$, and $\mathcal{P}_{pref}^m$, into the Integrator Agent. Formally, this process can be expressed as:

$$P_u^t \leftarrow \text{IAgent}(u, \{T_i^{\text{title}}, T_i^{\text{brand}}, T_i^{\text{categories}}, T_i^{\text{description}}\}_{i \in \mathcal{N}(u)}|\mathcal{P}_{pref}^t), \quad (1)$$

$$P_u^v \leftarrow \text{IAgent}(u, \{V_i\}_{i \in \mathcal{N}(u)}|\mathcal{P}_{pref}^v), \quad (2)$$

---

[2]https://github.com/enoche/MMRec

$$P_u^m \leftarrow \text{IAgent}(u, \{T_i^{\text{title}}, T_i^{\text{brand}}, T_i^{\text{categories}}, T_i^{\text{description}}, V_i\}_{i \in \mathcal{N}(u)} | \mathcal{P}_{pref}^m), \tag{3}$$

where $\mathcal{N}(u)$ denotes the interacted item set for user $u$. To ensure efficiency and account for the context length limitations of LLMs, for users $u$ who have interacted with more than a threshold number of items $\Upsilon$ ($|\mathcal{N}(u)| > \Upsilon$), we randomly select $\Upsilon$ items from interacted items. Users may exhibit specific preferences within individual modalities as well as preferences driven by cross-modal information. For instance, a user might simply prefer items of a certain color, favor items from a specific brand, or like items of a specific color from a particular brand. Therefore, we extract user preferences from three perspectives: textual, visual, and cross-modal with three different prompt templates (All templates are provided in Appendix D.1, Appendix D.2, and Appendix D.3 for details). All extracted user preferences are stored in the knowledge memory in a key-value format, allowing retrieval through the corresponding keys. Formally, memory format is expressed as:

$$\text{KMemory}(\text{Key} : (pref|u), \text{Value} : (P_u^t|P_u^v|P_u^m)). \tag{4}$$

Subsequently, we use pre-trained encoder $t_\theta(\cdot)$ to compute the representation of user preferences and construct a top-$k$ behavior- and multimodal-aware user-user graph $\mathcal{G}^{bmu} = (\mathcal{U}, \mathcal{E}^{bmu})$ based on cosine similarity. Formally, this process can be expressed as:

$$\mathcal{S}_{u,v}^{bmu} = \begin{cases} 1 & \text{if } \bar{\mathcal{S}}_{u,v}^{bmu} \in \text{top-}k\left(\bar{\mathcal{S}}_{u,*}^{bmu}\right) \\ 0 & \text{otherwise} \end{cases}, \quad \bar{\mathcal{S}}_{u,v}^{bmu} = \frac{t_\theta(P_u^t|P_u^v|P_u^m)^T t_\theta(P_v^t|P_v^v|P_v^m)}{\|t_\theta(P_u^t|P_u^v|P_u^m)\|\|t_\theta(P_v^t|P_v^v|P_v^m)\|}. \tag{5}$$

Then, we build unidirectional edges $(u, v) \in \mathcal{E}^{bmu}$, where $\mathcal{S}_{u,v}^{bmu} = 1$.

### 3.2.2 BEHAVIOR- AND MULTIMODAL-AWARE ITEM-ITEM GRAPH

Many existing multimodal recommendation models construct item-item graphs based on multimodal data. Our AgentMMRec also incorporates a refined traditional item-item graph (refer to Section 3.3). However, directly constructing an item-item graph using item features focuses only on the multimodal data itself, without considering the specific requirements of the recommendation task.

In recommendation systems, it is generally assumed that users who interact with the same items share similar preferences, and that items purchased by users with similar preferences exhibit similar properties. Therefore, for item $i$, the Integrator Agent leverages the powerful multimodal understanding and contextual processing capabilities of LLMs to integrate the multimodal data of other items purchased by users who have interacted with item $i$. This process enables the Integrator Agent to effectively generates behavior- and multimodal-aware properties $P_i$ for item $i$. In this process, tailored prompt templates $\mathcal{P}_{prop}^t$, $\mathcal{P}_{prop}^v$, and $\mathcal{P}_{prop}^m$ are fed into Integrator Agent for guidance. Formally, this process can be expressed as:

$$P_i^t \leftarrow \text{IAgent}(i, \{T_i^{\text{title}}, T_i^{\text{brand}}, T_i^{\text{categories}}, T_i^{\text{description}}\}_{i \in \mathcal{N}^2(i)} | \mathcal{P}_{prop}^t), \tag{6}$$

$$P_i^v \leftarrow \text{IAgent}(i, \{V_i\}_{i \in \mathcal{N}^2(i)} | \mathcal{P}_{prop}^v), \tag{7}$$

$$P_i^m \leftarrow \text{IAgent}(u, \{T_i^{\text{title}}, T_i^{\text{brand}}, T_i^{\text{categories}}, T_i^{\text{description}}, V_i\}_{i \in \mathcal{N}^2(i)} | \mathcal{P}_{prop}^m), \tag{8}$$

where $\mathcal{N}^2(i)$ denotes other items purchased by users who have interacted with item $i$. To ensure efficiency and account for the context length limitations of LLMs, for item set $\mathcal{N}^2(i)$ larger than a threshold number of items $\Upsilon$ ($|\mathcal{N}^2(i)| > \Upsilon$), we randomly select $\Upsilon$ items from $\mathcal{N}^2(i)$. For similar considerations as those in behavior- and multimodal-aware preferences, we extract item properties from three perspectives: textual, visual, and cross-modal, using three different prompt templates (All templates are provided in Appendix D.4, Appendix D.5, and Appendix D.6 for details). All extracted item properties are stored in the knowledge memory in a key-value format, enabling retrieval via the corresponding keys. Formally, the memory format is expressed as:

$$\text{KMemory}(\text{Key} : (prop|i), \text{Value} : (P_i^t|P_i^v|P_i^m)). \tag{9}$$

Subsequently, we use pre-trained encoder $t_\theta(\cdot)$ to compute the representation of item properties and construct a top-$k$ behavior- and multimodal-aware item-item graph $\mathcal{G}^{bmi} = (\mathcal{I}, \mathcal{E}^{bmi})$ based on cosine similarity. Formally, this process can be expressed as:

$$\mathcal{S}_{i,j}^{bmi} = \begin{cases} 1 & \text{if } \bar{\mathcal{S}}_{i,j}^{bmi} \in \text{top-}k\left(\bar{\mathcal{S}}_{i,*}^{bmi}\right) \\ 0 & \text{otherwise} \end{cases}, \quad \bar{\mathcal{S}}_{i,j}^{bmi} = \frac{t_\theta(P_i^t|P_i^v|P_i^m)^T t_\theta(P_j^t|P_j^v|P_j^m)}{\|t_\theta(P_i^t|P_i^v|P_i^m)\|\|t_\theta(P_j^t|P_j^v|P_j^m)\|}. \tag{10}$$

Then, we build unidirectional edges $(i, j) \in \mathcal{E}^{bmi}$, where $\mathcal{S}_{i,j}^{bmi} = 1$.

**Discussion.** The construction of behavior- and multimodal-aware homogeneous graphs is pre-built before training, eliminating any additional computational burden during the training process. Moreover, the stored knowledge can be continuously updated through feedback during training. Additionally, the threshold $\Upsilon$ further reduces computational overhead while considering the context length limitations of LLMs. A hyperparameter analysis of $\Upsilon$ is discussed in Appendix C.3.

## 3.3 REFINING TRADITIONAL ITEM-ITEM GRAPHS

Traditional multimodal recommendation models (Zhang et al., 2022; Xu et al., 2025c; Zhou & Shen, 2023) construct modality-specific item-item graphs based on item representations to enhance modality representations. However, this process exacerbates the isolation between modalities (Xu et al., 2025d) and lacks consideration of user preferences. Utilizer Agent leverages the behavior- and multimodal-aware preferences and properties stored in the constructed knowledge memory to refine and merge modality-specific item-item graphs into a unified item-item graph. Specifically, original modality-specific item-item graphs are constructed as:

$$\dagger\mathcal{S}_{i,j}^{i,m} = \begin{cases} 1 & \text{if } \dagger\bar{\mathcal{S}}_{i,j}^{i,m} \in \text{top-}k\left(\dagger\bar{\mathcal{S}}_{i,*}^{i,m}\right) \\ 0 & \text{otherwise} \end{cases}, \quad \dagger\bar{\mathcal{S}}_{i,j}^{i,m} = \frac{(\mathbf{e}_{i_m})^T \mathbf{e}_{j_m}}{\|\mathbf{e}_{i_m}\|\|\mathbf{e}_{j_m}\|}, \quad (11)$$

where $m \in t, v$. For each modality, we construct a top-$k$ modality-specific item-item graph. Then, for each item $i$, Utilizer Agent combines the multimodal data of the top-k items associated with item $i$ across all modalities to mitigate the isolation between modalities. Additionally, it extracts the behavior- and multimodal-aware properties of item $i$ and the behavior- and multimodal-aware preferences of users who have purchased item $i$ from the knowledge memory. Using a carefully designed prompt template $\mathcal{P}_{re}$ (Template is provided in Appendix D.7 for details), Utilizer Agent reselects the top-$k$ items for each item $i$, constructing a unified item-item graph $\mathcal{G}^{rti} = (\mathcal{I}, \mathcal{E}^{rti})$. Formally, this process can be expressed as:

$$\{\dagger\mathcal{S}_{i,*}^{rti}\}_{*\in\mathcal{I}} \leftarrow \text{UAgent}(i, \{\dagger\mathcal{S}_{i,*}^{i,m}\}_{*\in\mathcal{I}\&m\in t,v}, (P_i^t|P_i^v|P_i^m), \{(P_u^t|P_u^v|P_u^m)\}_{u\in\mathcal{N}(i)}|\mathcal{P}_{re}), \quad (12)$$

where $\mathcal{N}(i)$ denotes the purchased user set for item $i$ and number of selected items for each item $i$ is $k$ ($\sum_{i\in\mathcal{I}}\{\dagger\mathcal{S}_{i,*}^{rti}\}_{*\in\mathcal{I}} = k$). We also adopt $\Upsilon$ to constrain the size of the purchased user set $\mathcal{N}(i)$. Then, we build unidirectional edges $(i, j) \in \mathcal{E}^{rti}$, where $\dagger\mathcal{S}_{i,j}^{rti} = 1$.

**Discussion.** The refinement of modality-specific item-item graphs is also pre-conducted, adding no extra computational burden during training. The threshold $\Upsilon$ is also adopted to reduce overhead while addressing LLM context length limits. A hyperparameter analysis of $\Upsilon$ is discussed in Appendix C.3.

## 3.4 RERANKING FINAL ITEM LIST

We enhance user and item representations by leveraging encoded behavior- and multimodal-aware preferences and properties. Following the paradigm adopted by most previous studies (Xu et al., 2025f; Zhou et al., 2023a), we apply LightGCN (He et al., 2020) to propagate messages and perform readout over the user-item interaction graph $\mathcal{G}$. Subsequently, we enhance the user representations using the homogeneous graph $\mathcal{G}^{bmu}$, as in advanced multimodal recommendation models (Xu et al., 2025c; Wang et al., 2021), while item representations are enhanced using the homogeneous graphs $\mathcal{G}^{bmi}$ and $\mathcal{G}^{rti}$ (Xu et al., 2025c; Zhou & Shen, 2023). The model is optimized using BPR loss function (Rendle et al., 2009). Since graph-based multimodal recommendation paradigms are relatively mature, we provide a detailed introduction in Appendix A.1. Additionally, AgentMMRec can benefit from more sophisticated self-supervised tasks (Xu et al., 2025e; Zhou et al., 2023b; Wei et al., 2023). For efficiency considerations, we did not incorporate any self-supervised tasks but included related experiments in Appendix C.5.

For the recommendation stage, Utilizer Agent reranks the final item list for each user $u$ by combining the behavior- and multimodal-aware preferences and properties of user $u$ and the items in the list. This process, under the guidance of a tailored prompt template $\mathcal{P}_{rank}$, can be expressed as:

$$\bar{\mathcal{I}}_u \leftarrow \text{UAgent}(u, \mathcal{I}_u, (P_u^t|P_u^v|P_u^m), \{(P_i^t|P_i^v|P_i^m)\}_{i\in\mathcal{I}_u}|\mathcal{P}_{rank}), \quad (13)$$

where $\mathcal{I}_u$ denotes final item list for user $u$. We use single-item NDCG@$N$ as the evaluation metric to determine whether rerankings produce a positive effect, where $N = |\mathcal{I}_u|$. We define $\text{Eval}(u, \mathcal{I}_u)$ as NDCG@$N$ performance of user $u$'s final item list $\mathcal{I}_u$. If $\text{Eval}(u, \mathcal{I}_u) > \text{Eval}(u, \bar{\mathcal{I}}_u)$, it indicates that the reranking has produced a negative effect. In cases where rerankings produce a negative effect, Integrator Agent leverages the multimodal data from the ground-truth item list $\hat{\mathcal{I}}_u$ and user $u$'s existing behavior- and multimodal-aware preferences to updates the knowledge store to refine user $u$'s preferences under the guidance of a tailored prompt template $\mathcal{P}_{update}$. Formally:

$$(P_u^t|P_u^v|P_u^m) \leftarrow \text{IAgent}(u, \hat{\mathcal{I}}_u, (P_u^t|P_u^v|P_u^m), \{(P_i^t|P_i^v|P_i^m)\}_{i \in \hat{\mathcal{I}}_u}|\mathcal{P}_{update}). \tag{14}$$

After updating $u$'s behavior- and multimodal-aware preferences, the process iteratively reranks and evaluates $u$'s final item list until the reranking produces a positive effect. Once a positive effect is achieved, the loop stops, and the knowledge memory is updated accordingly. For efficiency considerations, we perform knowledge updates every $E$ epochs. Templates $\mathcal{P}_{rank}$ and $\mathcal{P}_{update}$ are provided in Appendix D.8 and Appendix D.9 for details.

## 4 EXPERIMENT

### 4.1 EXPERIMENT SETUP

**Datasets.** The experiments are conducted on three real-world datasets containing two modalities: Baby, Sports, and Clothing from the Amazon dataset (McAuley et al., 2015). These datasets include textual and visual features, derived from item descriptions and corresponding images. The data preprocessing for these datasets follows the methodology outlined in MMRec (Zhou, 2023). Table 3 in Appendix B.1 shows the statistics of these datasets.

**Metrics.** For a fair comparison, we follow the settings of previous works (Xu et al., 2025f; Zhou et al., 2023b; Zhou & Shen, 2023) to adopt two widely-used evaluation metrics for top-$N$ recommendation: Recall@$N$ and NDCG@$N$. We report the average scores for all users in the test dataset under both $N = 10$ and $N = 20$, respectively.

**Baselines.** To evaluate the effectiveness of AgentMMRec, we compare it with the following baselines, including **MMGCN** (Wei et al., 2019), **DualGNN** (Wang et al., 2021), **LATTICE** (Zhang et al., 2022), **SLMRec** (Tao et al., 2022), **FREEDOM** (Zhou & Shen, 2023), **BM3** (Zhou et al., 2023b), **MMSSL** (Wei et al., 2023), **LLMRec** (Wei et al., 2024), **LGMRec** (Guo et al., 2024), **DiffMM** (Jiang et al., 2024), **SMORE** (Ong & Khong, 2025), **BeFA** (Fan et al., 2025), **MENTOR** (Xu et al., 2025e), **COHESION** (Xu et al., 2025c), **HPMRec** (Chen et al., 2025b), EVEN (Qi et al., 2025) and FreRec (Peng et al., 2025). Details can be found in Appendix B.2.

**Implementation Details.** We retain the standard settings for all baselines and fix batch size as 2048. For each of the selected baselines, the hyperparameters were tuned in line with the optimal configurations reported in the respective published papers. All baselines are implemented in PyTorch, using the Adam optimizer (Kingma & Ba, 2014) and Xavier initialization (Glorot & Bengio, 2010) with default parameters. To ensure fairness, we use the pre-trained text and vision encoders $t_\theta(\cdot)$ and $v_\theta(\cdot)$ provided by MMRec Framework (Xu et al., 2025f). For Integrator Agent and Utilizer Agent in AgentMMRec, we choose Qwen2.5-VL-7B. In Appendix C.4, we further explore whether larger parameter version (Qwen2.5-VL-32B) or powerful LLM (GPT-4o) provide additional advantages. For efficiency considerations, we set $E = 10$ for knowledge update.

### 4.2 OVERALL PERFORMANCE

We evaluate the effectiveness of AgentMMRec on multiple real-world datasets in multimodal recommendation scenarios. From Table 1, we find the following observations:

- 1. AgentMMRec achieves significant performance improvements over all baselines across datasets, demonstrating its effectiveness in bridging the semantic gap between multimodal data and recommendation tasks. This success stems from the synergistic roles of the Integrator Agent and Utilizer Agent. Integrator Agent harnesses the multimodal understanding and contextual processing capabilities of LLMs to infer behavior- and multimodal-aware user preferences and item properties from historical interactions and multimodal item information, constructing effective homogeneous

Table 1: Performance comparison of baselines and AgentMMRec in terms of Recall and NDCG. * indicates that the t-tests validate the significance of performance improvements with $p$-value $< 0.05$.

| Datasets | Baby | | | | Sports | | | | Clothing | | | |
|---|---|---|---|---|---|---|---|---|---|---|---|---|
| Metrics | R@10 | R@20 | N@10 | N@20 | R@10 | R@20 | N@10 | N@20 | R@10 | R@20 | N@10 | N@20 |
| MMGCN (MM'19) | 0.0378 | 0.0615 | 0.0200 | 0.0261 | 0.0370 | 0.0605 | 0.0193 | 0.0254 | 0.0218 | 0.0345 | 0.0110 | 0.0142 |
| DualGNN (TMM'21) | 0.0448 | 0.0716 | 0.0240 | 0.0309 | 0.0568 | 0.0859 | 0.0310 | 0.0385 | 0.0454 | 0.0683 | 0.0241 | 0.0299 |
| LATTICE (MM'21) | 0.0547 | 0.0850 | 0.0292 | 0.0370 | 0.0620 | 0.0953 | 0.0335 | 0.0421 | 0.0492 | 0.0733 | 0.0268 | 0.0330 |
| SLMRec (TMM'22) | 0.0529 | 0.0775 | 0.0290 | 0.0353 | 0.0663 | 0.0990 | 0.0365 | 0.0450 | 0.0452 | 0.0675 | 0.0247 | 0.0303 |
| FREEDOM (MM'23) | 0.0627 | 0.0992 | 0.0330 | 0.0424 | 0.0717 | 0.1089 | 0.0385 | 0.0481 | 0.0629 | 0.0941 | 0.0341 | 0.0420 |
| BM3 (WWW'23) | 0.0564 | 0.0883 | 0.0301 | 0.0383 | 0.0656 | 0.0980 | 0.0355 | 0.0438 | 0.0422 | 0.0621 | 0.0231 | 0.0281 |
| MMSSL (WWW'23) | 0.0613 | 0.0971 | 0.0326 | 0.0420 | 0.0693 | 0.1013 | 0.0369 | 0.0474 | 0.0531 | 0.0797 | 0.0291 | 0.0359 |
| LLMRec (WSDM'24) | 0.0621 | 0.0983 | 0.0324 | 0.0422 | 0.0682 | 0.1000 | 0.0363 | 0.0459 | 0.0540 | 0.0808 | 0.0294 | 0.0365 |
| LGMRec (AAAI'24) | 0.0639 | 0.0989 | 0.0337 | 0.0430 | 0.0719 | 0.1068 | 0.0387 | 0.0477 | 0.0555 | 0.0828 | 0.0302 | 0.0371 |
| DiffMM (MM'24) | 0.0623 | 0.0975 | 0.0328 | 0.0411 | 0.0671 | 0.1017 | 0.0377 | 0.0458 | 0.0531 | 0.0797 | 0.0291 | 0.0359 |
| SMORE (WSDM'25) | 0.0680 | 0.1035 | 0.0365 | 0.0457 | 0.0762 | 0.1142 | 0.0408 | 0.0506 | 0.0659 | 0.0987 | 0.0360 | 0.0443 |
| FreRec (MM'25) | 0.0662 | 0.1011 | 0.0348 | 0.0437 | 0.0754 | 0.1147 | 0.0410 | 0.0508 | 0.0674 | 0.0977 | 0.0363 | 0.0447 |
| EVEN (AAAI'25) | 0.0667 | 0.1031 | 0.0355 | 0.0448 | 0.0759 | 0.1143 | 0.0411 | 0.0510 | 0.0662 | 0.0978 | 0.0356 | 0.0436 |
| BeFA (AAAI'25) | 0.0555 | 0.0884 | 0.0299 | 0.0383 | 0.0649 | 0.0985 | 0.0346 | 0.0432 | 0.0568 | 0.0857 | 0.0307 | 0.0381 |
| MENTOR (AAAI'25) | 0.0678 | 0.1048 | 0.0362 | 0.0450 | 0.0763 | 0.1139 | 0.0409 | 0.0511 | 0.0668 | 0.0989 | 0.0360 | 0.0441 |
| COHESION (SIGIR'25) | 0.0680 | 0.1052 | 0.0354 | 0.0454 | 0.0752 | 0.1137 | 0.0409 | 0.0503 | 0.0665 | 0.0983 | 0.0358 | 0.0438 |
| HPMRec (CIKM'25) | 0.0667 | 0.1033 | 0.0357 | 0.0451 | 0.0751 | 0.1129 | 0.0410 | 0.0507 | 0.0658 | 0.0963 | 0.0351 | 0.0429 |
| AgentMMRec (Qwen) | **0.0705*** | **0.1079*** | **0.0380*** | **0.0475*** | **0.0838*** | **0.1231*** | **0.0454*** | **0.0557*** | **0.0740*** | **0.1071*** | **0.0404*** | **0.0490*** |

graphs. Utilizer Agent then refines traditional item-item graphs and reranks the final item list using these enriched user preferences and item properties. Additionally, Integrator Agent updates behavior- and multimodal-aware user preferences based on the evaluation of the reranked results. In Section 4.3, we validate the effectiveness of each component through detailed ablation studies.

- 2. Suboptimal baselines (SMORE, MENTOR, COHESION, and HPMRec) exhibit similar performance despite their differing designs. For example, SMORE employs spectral fusion, MENTOR utilizes tailored modality alignment, COHESION constructs composite graphs, and HPMRec applies hypercomplex operators. However, all encounter a consistent performance bottleneck, which we attribute to the inherent semantic gaps between multimodal data and recommendation tasks, as well as knowledge limitations. To test this hypothesis, in Section 4.4, we transfer the behavior- and multimodal-aware homogeneous graphs constructed by AgentMMRec to these baselines or allow Utilizer Agent to leverage AgentMMRec's optimized knowledge memory to rerank their outputs, aiming to overcome their bottlenecks.

## 4.3 ABLATION STUDY

To validate the effectiveness of AgentMMRec, we conduct experiments to justify the importance of key components. We design following variants: (1) $w/o$ bmh, which removes both behavior- and multimodal-aware user-user and item-item graphs. (2) $w/o$ bmu, which removes behavior- and multimodal-aware user-user graph. (3) $w/o$ bmi, which removes behavior- and

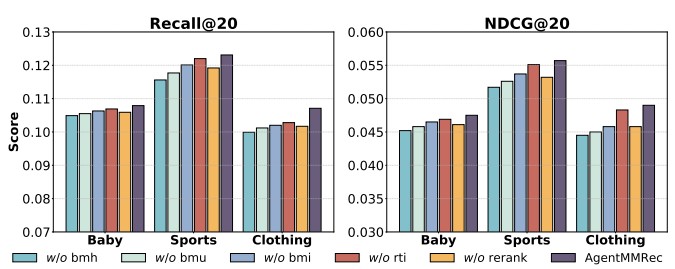

Figure 2: Ablation study for AgentMMRec across all datasets.

multimodal-aware item-item graph. (4) $w/o$ rti, which directly uses traditional item-item graphs to replace unified item-item graph. (5) $w/o$ rerank, which removes rerank and feedback process. Notably, for variants (1)-(3), Integrator Agent still extracts and stores behavior- and multimodal-aware user preferences and item properties. Figure 2 shows that each component contributes to the performance improvement of AgentMMRec. In Section 4.4, we further explore whether the key components of AgentMMRec can be transferred to existing models to break through their performance bottlenecks.

## 4.4 COMPATIBILITY ANALYSIS

We conducted two distinct compatibility experiments: (1) transferring the behavior- and multimodal-aware homogeneous graphs constructed by AgentMMRec to the suboptimal baselines and (2) allowing the Utilizer Agent to leverage the knowledge memory optimized by AgentMMRec to rerank the suboptimal baselines. We select suboptimal baselines (SMORE,MENTOR,COHESION, and HPM-Rec) in Table 1. Two variants represented as (1) $+Graph$ and (2) $+Rerank$. For $+Graph$ variant,

Table 2: Compatibility analysis of AgentMMRec with suboptimal baselines.

| Models | Datasets | Baby | | | | Sports | | | | Clothing | | | |
|---|---|---|---|---|---|---|---|---|---|---|---|---|---|
| | Metrics | R@10 | R@20 | N@10 | N@20 | R@10 | R@20 | N@10 | N@20 | R@10 | R@20 | N@10 | N@20 |
| SMORE | Original | 0.0680 | 0.1035 | 0.0365 | 0.0457 | 0.0762 | 0.1142 | 0.0408 | 0.0506 | 0.0659 | 0.0987 | 0.0360 | 0.0443 |
| | +Graph | **0.0691** | **0.1055** | **0.0371** | **0.0466** | **0.0799** | **0.1190** | **0.0437** | **0.0532** | **0.0709** | **0.1041** | **0.0388** | **0.0473** |
| | +Rerank | 0.0686 | 0.1047 | 0.0369 | 0.0463 | 0.0785 | 0.1170 | 0.0431 | 0.0527 | 0.0688 | 0.1021 | 0.0377 | 0.0462 |
| MENTOR | Original | 0.0678 | 0.1048 | 0.0362 | 0.0450 | 0.0763 | 0.1139 | 0.0409 | 0.0511 | 0.0668 | 0.0989 | 0.0360 | 0.0441 |
| | +Graph | **0.0693** | **0.1061** | **0.0370** | **0.0461** | **0.0792** | **0.1180** | **0.0434** | **0.0532** | **0.0707** | **0.1035** | **0.0383** | **0.0466** |
| | +Rerank | 0.0685 | 0.1053 | 0.0366 | 0.0453 | 0.0778 | 0.1164 | 0.0427 | 0.0528 | 0.0689 | 0.1024 | 0.0370 | 0.0455 |
| COHESION | Original | 0.0680 | 0.1052 | 0.0354 | 0.0454 | 0.0752 | 0.1137 | 0.0409 | 0.0503 | 0.0665 | 0.0983 | 0.0358 | 0.0438 |
| | +Graph | **0.0695** | **0.1066** | **0.0365** | **0.0460** | **0.0780** | **0.1174** | **0.0430** | **0.0525** | **0.0697** | **0.1033** | **0.0380** | **0.0462** |
| | +Rerank | 0.0686 | 0.1059 | 0.0358 | 0.0458 | 0.0773 | 0.1159 | 0.0423 | 0.0519 | 0.0681 | 0.1015 | 0.0370 | 0.0453 |
| HPMRec | Original | 0.0667 | 0.1033 | 0.0357 | 0.0451 | 0.0751 | 0.1129 | 0.0410 | 0.0507 | 0.0658 | 0.0963 | 0.0351 | 0.0429 |
| | +Graph | **0.0682** | **0.1054** | **0.0366** | **0.0459** | **0.0785** | **0.1174** | **0.0432** | **0.0530** | **0.0698** | **0.1025** | **0.0375** | **0.0449** |
| | +Rerank | 0.0677 | 0.1042 | 0.0360 | 0.0454 | 0.0776 | 0.1162 | 0.0427 | 0.0525 | 0.0684 | 0.1004 | 0.0362 | 0.0440 |

we follow previous studies (Zhou & Shen, 2023; Xu et al., 2025c) to conduct graph convolution operation. Results in Table 2 verifies that the performance bottlenecks of suboptimal baselines are constrained by the semantic gap between multimodal information and recommendation tasks and knowledge limitations. Additionally, it also demonstrates that AgentMMRec can effectively bridge the semantic gap and provide enriched knowledge.

## 4.5 SPARSITY ANALYSIS

We evaluate the effectiveness of AgentMMRec across varying levels of data sparsity. To assess its performance, we conduct experiments on sub-datasets derived from all datasets, each exhibiting different degrees of sparsity. AgentMMRec is compared against five competitive baselines: LGMRec, SMORE, MENTOR, COHESION, and HPMRec. Users are categorized into groups based on the number of interactions in the training set, such as those with $0 - 5$ interacted items in the first group. As shown in Figure 3, AgentMMRec consistently outperforms all baselines across datasets, demonstrating its robustness under varying levels of sparsity.

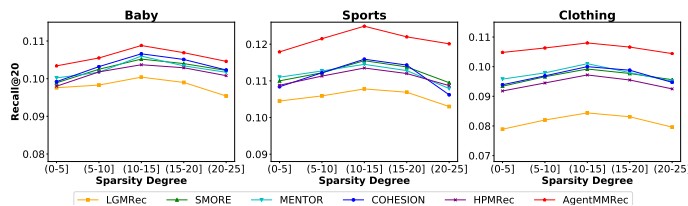

Figure 3: Sparsity analysis for AgentMMRec across all datasets.

## 4.6 IN-DEPTH ANALYSIS

Due to space limitations, we provide an in-depth analysis in the appendix. Specifically, we explore AgentMMRec's performance in cold-start scenario in Appendix C.1. The analysis and discussion of hyperparameters can be found in Appendix C.3, while the discussion on replacing the LLM backbone for agents included in Appendix C.4. Furthermore, the potential benefits of popular modality-alignment self-supervised tasks to AgentMMRec are discussed in Appendix C.5. Moreover, we further explore whether the knowledge memory can benefit from multiple rounds of updating with AgentMMRec or relay updating with multiple different models, which is detailed in Appendix C.2.

## 5 CONCLUSION

In this paper, we identify that current multimodal recommendations are hindered by the semantic gap between multimodal data and recommendation tasks. Leveraging the multimodal understanding and contextual processing capabilities of LLMs, we propose AgentMMRec, a novel agent-based framework that effectively bridges this semantic gap through two cooperative agents. These agents achieve this by constructing behavior- and multimodal-aware homogeneous graphs, refining traditional item-item graphs, reranking the final item list, and updating the knowledge memory. Extensive experiments demonstrate that AgentMMRec achieves significant performance improvements and excels under various data sparsity scenarios. Furthermore, AgentMMRec has the ability to integrate with existing models to overcome their performance bottlenecks.

This paper also offers a promising future direction: shifting the focus from solely model design to exploring the fundamental relationship between data and tasks.

## 6  ETHICS STATEMENT

Our work adheres to the ethical guidelines outlined in the ICLR Code of Ethics.

## 7  REPRODUCIBILITY STATEMENT

The code is available at the anonymous repository link listed at the end of the abstract. The detailed experimental setup, in-depth experiments, and all prompt templates are thoroughly described in the appendix.

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

# Appendix

# A  Technique Details

## A.1  Technique Details before Reranking

We provide technique details prior to reranking. First, we utilize LightGCN (He et al., 2020) to extract high-order user-item collaborative signals, formally the embeddings for user $u$ and item $i$ in the $l-$th layer are:

$$\hat{\mathbf{e}}_u^{(l)} = \frac{1}{\mathcal{N}(u)} \sum_{j|(u,j)\in\mathcal{E}} \frac{1}{\mathcal{N}(j)} \hat{\mathbf{e}}_j^{(l-1)}, \quad \hat{\mathbf{e}}_i^{(l)} = \frac{1}{\mathcal{N}(i)} \sum_{v|(i,v)\in\mathcal{E}} \frac{1}{\mathcal{N}(v)} \hat{\mathbf{e}}_v^{(l-1)}, \quad (15)$$

where $\hat{\mathbf{e}}_u = (\mathbf{e}_{u_t}|\mathbf{e}_{u_v}|t_\theta((P_u^t|P_u^v|P_u^m)))$ and $\hat{\mathbf{e}}_i = (\mathbf{e}_{i_t}|\mathbf{e}_{i_v}|t_\theta((P_i^t|P_i^v|P_i^m)))$ are user and item representations enhanced by extracted behavior- and multimodal-aware user preferences and item properties. Here $|$ denote concatenation operation. After $L$ layers of graph convolution operation, the final representations of user $u$ and item $i$ are calculated as:

$$\hat{\mathbf{e}}_u = \sum_{l=0}^{L} \hat{\mathbf{e}}_u^l, \quad \hat{\mathbf{e}}_i = \sum_{l=0}^{L} \hat{\mathbf{e}}_i^l. \quad (16)$$

Here, we fix $L = 3$ for all experiments, which is the best setting in most multimodal recommendation models (Xu et al., 2025f). Entire user and item representations can be formulated as $\hat{\mathbf{E}}_u$ and $\hat{\mathbf{E}}_i$, respectively. Furthermore, we adopts constructed behavior- and multimodal-aware homogeneous graphs and refined unified item-item graph to enhance user and item representations.

For user side, we only have constructed behavior- and multimodal-aware user-user graph $\mathcal{E}^{bmu}$ with similarity matrix $\mathcal{S}^{bmu}$. Therefore, user side representation enhancement can be expressed as:

$$\bar{\mathbf{E}}_u = \hat{\mathbf{E}}_u + \hat{\mathbf{E}}_u((\mathcal{D}^{bmu})^{-\frac{1}{2}}\mathcal{S}^{bmu}(\mathcal{D}^{bmu})^{-\frac{1}{2}}), \quad (17)$$

where $\mathcal{D}^{bmu}$ is the diagonal degree matrix of $\mathcal{S}^{bmu}$. This normalization aim to mitigate the issues of gradient explosion or vanishing.

For item side, we have constructed behavior- and multimodal-aware item-item graph $\mathcal{E}^{bmi}$ with similarity matrix $\mathcal{S}^{bmi}$ and refined unified item-item graph $\mathcal{E}^{rti}$ with similarity matrix $\dagger\mathcal{S}^{rti}$. Therefore, item side representation enhancement can be expressed as:

$$\bar{\mathbf{E}}_i = \hat{\mathbf{E}}_i + \hat{\mathbf{E}}_i((\mathcal{D}^{bmi})^{-\frac{1}{2}}\mathcal{S}^{bmi}(\mathcal{D}^{bmi})^{-\frac{1}{2}}) + \hat{\mathbf{E}}_i((\mathcal{D}^{rti})^{-\frac{1}{2}}\dagger\mathcal{S}^{rti}(\mathcal{D}^{rti})^{-\frac{1}{2}}), \quad (18)$$

where $\mathcal{D}^{bmi}$ and $\mathcal{D}^{rti}$ are the diagonal degree matrices of $\mathcal{S}^{bmi}$ and $\dagger\mathcal{S}^{rti}$, respectively. These normalizations also aim to mitigate the issues of gradient explosion or vanishing.

Notably, for efficiency consideration, all homogeneous graph only adopt single-layer convolution.

Consistent with almost all existing multimodal recommendation studies (Xu et al., 2025f; Liu et al., 2024; Zhou et al., 2023a), we use BPR for model optimization. Specifically, we compute the inner product of user and item representations to calculate predicted scores and adopt the BPR loss function:

$$\mathcal{L}_{bpr} = \sum_{(u,p,n)\in\mathcal{D}} -\log\left(\sigma\left(\bar{\mathbf{e}}_u^\top\bar{\mathbf{e}}_p - \bar{\mathbf{e}}_u^T\bar{\mathbf{e}}_n\right)\right), \quad (19)$$

where $\sigma(\cdot)$ is the Sigmoid function. $p$ and $n$ denote positive and negative items for user $u$, respectively.

## A.2  Algorithm

We provide algorithmic pseudocode in Algorithm 1 to provide overview of our AgentMMRec.

# B  Experimental Settings

## B.1  Datasets

Each dataset was preprocessed using a 5-core filtering setting to eliminate infrequent users and items. The statistical characteristics of the filtered datasets are summarized in Table 3. The processed data were then split into training, validation, and test sets in an 8:1:1 ratio.

---

**Algorithm 1** Process of AgentMMRec

---

1: **Input:** User set $\mathcal{U}$, item set $\mathcal{I}$, item textual data (Title: $T_i^{title}$, Brand: $T_i^{brand}$, Categories: $T_i^{categories}$, and Description: $T_i^{description}$), item visual data (Image: $V_i$), pretrained textual encoder $t_\theta(\cdot)$, pretrained visual encoder $v_\theta(\cdot)$, user-item graph $\mathcal{G}$, Integrator Agent IAgent$(\cdot)$, Utilizer Agent UAgent$(\cdot)$, knowledge memory KMemory$(\cdot)$, prompt templates ($\{\mathcal{P}_{pref}^*\}_{*\in t,v,m}$, $\{\mathcal{P}_{prop}^*\}_{*\in t,v,m}$, $\mathcal{P}_{re}$, $\mathcal{P}_{rank}$, and $\mathcal{P}_{update}$), and knowledge useful flag $f$;

2: Extract item representations $\mathbf{E}_{i_t}$, $\mathbf{E}_{i_v}$ textual and visual modalities via encoder $t_\theta(\cdot)$ and $v_\theta(\cdot)$;

3: Randomly initialize user representations $\mathbf{E}_{u_t}$, $\mathbf{E}_{u_v}$;

4: Generate behavior- and multimodal-aware user preferences $(P_u^t, P_u^v, P_u^m)$ for each user $u$ via Integrator Agent IAgent$(\cdot)$, item textual data (Title: $T_i^{title}$, Brand: $T_i^{brand}$, Categories: $T_i^{categories}$, and Description: $T_i^{description}$), item visual data (Image: $V_i$), and prompt templates $\{\mathcal{P}_{pref}^*\}_{*\in t,v,m}$;

5: Memory user preferences $(P_u^t, P_u^v, P_u^m)$ into knowledge memory KMemory$(\cdot)$ for each user $u$;

6: Construct behavior- and multimodal-aware user-user graph $\mathcal{G}^{bmu}$ via behavior- and multimodal-aware user preferences $(P_u^t, P_u^v, P_u^m)$ for each user $u$ and pretrained textual encoder $t_\theta(\cdot)$.

7: Generate behavior- and multimodal-aware item properties $(P_i^t, P_i^v, P_i^m)$ for each item $i$ via Integrator Agent IAgent$(\cdot)$, item textual data (Title: $T_i^{title}$, Brand: $T_i^{brand}$, Categories: $T_i^{categories}$, and Description: $T_i^{description}$), item visual data (Image: $V_i$), and prompt templates $\{\mathcal{P}_{prop}^*\}_{*\in t,v,m}$;

8: Memory item properties $(P_i^t, P_i^v, P_i^m)$ into knowledge memory KMemory$(\cdot)$ for each item $i$;

9: Construct behavior- and multimodal-aware item-item graph $\mathcal{G}^{bmi}$ via behavior- and multimodal-aware item properties $(P_i^t, P_i^v, P_i^m)$ for each item $i$ and pretrained textual encoder $t_\theta(\cdot)$.

10: Build traditional modality-specific item-item graph $\dagger\mathcal{S}_{i,j}^{i,m}$ for each modality $m$ via $\mathbf{E}_{i_t}$ and $\mathbf{E}_{i_v}$;

11: Refine and construct unified item-item graph $\mathcal{G}^{rti}$ via traditional modality-specific item-item graph $\{\dagger\mathcal{S}_{i,j}^{i,m}\}_{m\in t,v}$, user preferences $(P_u^t, P_u^v, P_u^m)$ for each user $u$, item properties $(P_i^t, P_i^v, P_i^m)$ for each item $i$, and prompt template $\mathcal{P}_{re}$;

12: **while** not converged **do**

13:   Enhance user representations $\hat{\mathbf{E}}_u$ via entire user representations $(\mathbf{E}_{u_t}, \mathbf{E}_{u_v})$, user preferences $(P_u^t, P_u^v, P_u^m)$ for each user $u$, and textual encoder $t_\theta(\cdot)$;

14:   Enhance item representations $\hat{\mathbf{E}}_i$ via entire item representations $(\mathbf{E}_{i_t}, \mathbf{E}_{i_v})$, item preferences $(P_i^t, P_i^v, P_i^m)$ for each item $i$, and textual encoder $t_\theta(\cdot)$;

15:   Extract high-order user-item collaborative signals via user-item graph $\mathcal{G}$, user representations $\hat{\mathbf{E}}_u$, and item representations $\hat{\mathbf{E}}_i$;

16:   Get enhanced user representations $\bar{\mathbf{E}}_u$ via user representations $\hat{\mathbf{E}}_u$ and behavior- and multimodal-aware user-user graph $\mathcal{G}^{bmu}$;

17:   Get enhanced user representations $\bar{\mathbf{E}}_i$ via user representations $\hat{\mathbf{E}}_i$, behavior- and multimodal-aware item-item graph $\mathcal{G}^{bmi}$, and unified item-item graph $\mathcal{G}^{rti}$;

18:   Optimize model via BPR loss function and get final item list $\mathcal{I}_u$ for each user $u$;

19:   Set knowledge useful flag $f$ = False;

20:   **while** $f$ = False **do**

21:     Rerank final item list $\bar{\mathcal{I}}_u$ for each user $u$ via Utilizer Agent UAgent, user preferences $(P_u^t, P_u^v, P_u^m)$ for each user $u$, item preferences $(P_i^t, P_i^v, P_i^m)$ for each item $i$, and prompt template $\mathcal{P}_{rank}$;

22:     Evaluate final item list $\mathcal{I}_u$ for each user $u$ (Eval$(u, \mathcal{I}_u)$), and reranked final item list $\bar{\mathcal{I}}_u$ for each user $u$ (Eval$(u, \bar{\mathcal{I}}_u)$);

23:     **if** Eval$(u, \mathcal{I}_u)$ < Eval$(u, \bar{\mathcal{I}}_u)$ **then**

24:       Set knowledge useful flag $f$ = TRUE;

25:     **else**

26:       Update user preferences $(P_u^t, P_u^v, P_u^m)$ for each user $u$ via Integrator Agent IAgent$(\cdot)$, user preferences $(P_u^t, P_u^v, P_u^m)$ for each user $u$, item preferences $(P_i^t, P_i^v, P_i^m)$ for each item $i$, ground-truth item list $\hat{\mathcal{I}}_u$ for each user $u$, and prompt template $\mathcal{P}_{update}$;

27:     **end if**

28:   **end while**

29: **end while**

---

Table 3: Statistics of all evaluation datasets.

| Datasets | #Users | #Items | #Interactions | Sparsity |
|---|---|---|---|---|
| Baby | 19,445 | 7,050 | 160,792 | 99.88% |
| Sports | 35,598 | 18,357 | 296,337 | 99.95% |
| Clothing | 39,387 | 23,033 | 278,677 | 99.97% |

## B.2 BASELINES

To evaluate the effectiveness of AgentMMRec, we compare it with the following baselines, including **MMGCN** (Wei et al., 2019), **DualGNN** (Wang et al., 2021), **LATTICE** (Zhang et al., 2022), **SLMRec** (Tao et al., 2022), **FREEDOM** (Zhou & Shen, 2023), **BM3** (Zhou et al., 2023b), **MMSSL** (Wei et al., 2023), **LLMRec** (Wei et al., 2024), **LGMRec** (Guo et al., 2024), **DiffMM** (Jiang et al., 2024), **SMORE** (Ong & Khong, 2025), **BeFA** (Fan et al., 2025), **MENTOR** (Xu et al., 2025e), **COHESION** (Xu et al., 2025c), and **HPMRec** (Chen et al., 2025b). Specifically:

- **MMGCN** (Wei et al., 2019): It processes and integrates information of different modalities through graph convolutional networks (GCN).

- **DualGNN** (Wang et al., 2021): It combines multi-modal information of users and items. It builds an extra user-user graph to capture user behavior to improve recommendation quality.

- **LATTICE** (Zhang et al., 2021): It constructs an extra item semantic graph to capture the latent semantically correlative signals.

- **SLMRec** (Tao et al., 2022): It utilizes self-supervised learning (SSL) for multimodal recommendation, improving recommendation performance through noise perturbation of features and multi-modal pattern uncovering enhancement tasks.

- **FREEDOM** (Zhou & Shen, 2023): It denoises the user-item graph and builds a frozen item-item graph through original modality features to improve recommendation performance.

- **BM3** (Zhou et al., 2023b): It implifies the SSL task for multimodal recommendation. It utilizes the dropout mechanism to perturb the representation.

- **MMSSL** (Wei et al., 2023): It designs a modality-aware adversarial perturbation-based interactive structure learning paradigm and proposes a cross-modal comparative learning method to distinguish common features and specific features between modalities.

- **LLMRec** (Wei et al., 2024): It employs several effective LLM-based graph augmentation strategies to enhance recommendation performance.

- **LGMRec** (Guo et al., 2024): It captures and utilizes local topological information and global embeddings with hypergraph structure.

- **DiffMM** (Jiang et al., 2024): It integrates a modality-aware graph diffusion model with a cross-modal contrastive learning paradigm to improve modality-aware user representation learning.

- **SMORE** (Ong & Khong, 2025): It reduces modality noise by harnessing the discriminative spectrum property and global perspective inherent in the frequency domain.

- **BeFA** (Fan et al., 2025): It corrects multimodal features based on user behavior.

- **MENTOR** (Xu et al., 2025e): It proposes multi-level cross-modal alignment tasks to effectively improve final representation and achieve state-of-the-art recommendation accuracy.

- **COHESION** (Xu et al., 2025c): It introduces a tailored dual-stage fusion mecanism to liberate the effectiveness of composite graphs.

- **HPMRec** (Chen et al., 2025b): It enriches feature diversity and bridges semantic gaps across modalities.

- EVEN (Qi et al., 2025): It evaluates and denoises information in multimodal content and observed interactions.

- FreRec (Peng et al., 2025): It addresses modality noise and limited fusion in multimodal recommendation.

## C  IN-DEPTH ANALYSIS

### C.1  COLD-START ANALYSIS

We present results on the item cold-start scenario across all datasets (following widely used settings (Zhang et al., 2022), (Xu et al., 2025f)). The experimental results in Table 4 show that AgentMMRec significantly outperforms all baselines in the cold-start scenario. We attribute this to AgentMMRec's ability to fully leverage multimodal information, enabling accurate identification of properties for new items, thereby enhancing the alignment between multimodal data and recommendation tasks. Moreover, based on the experimental results and model design, we provide some insights into item cold-start. We observe that models such as FREEDOM, LLMRec, LGMRec, SMORE, MENTOR, COHESION, HPMRec, and AgentMMRec, which construct an item-item graph, have a significant impact on improving item cold-start. This indicates that multimodal data can effectively capture and reflect item properties. The advantage of AgentMMRec lies partly in the multimodal data provided by LLMs, which incorporates user behavior and aligns more closely with the recommendation task. Notably, in multimodal recommendation scenarios, new items come with multimodal data, allowing the model to effectively extract the properties of cold-start items. However, since new users lack interaction records and personal profiles, cold-start users are difficult to explore.

Table 4: Item cold-start analysis across all datasets.

| Datasets | Baby | | | | Sports | | | | Clothing | | | |
|---|---|---|---|---|---|---|---|---|---|---|---|---|
| Metrics | R@10 | R@20 | N@10 | N@20 | R@10 | R@20 | N@10 | N@20 | R@10 | R@20 | N@10 | N@20 |
| MMGCN | 0.0103 | 0.0186 | 0.0062 | 0.0098 | 0.0106 | 0.0178 | 0.0060 | 0.0094 | 0.0069 | 0.0101 | 0.0039 | 0.0050 |
| DualGNN | 0.0132 | 0.0200 | 0.0077 | 0.0110 | 0.0166 | 0.0242 | 0.0096 | 0.0132 | 0.0134 | 0.0198 | 0.0082 | 0.0113 |
| LATTICE | 0.0175 | 0.0256 | 0.0099 | 0.0138 | 0.0266 | 0.0340 | 0.0139 | 0.0189 | 0.0168 | 0.0249 | 0.0095 | 0.0135 |
| SLMRec | 0.0172 | 0.0259 | 0.0101 | 0.0140 | 0.0281 | 0.0354 | 0.0142 | 0.0194 | 0.0208 | 0.0208 | 0.0084 | 0.0118 |
| FREEDOM | 0.0348 | 0.0588 | 0.0195 | 0.0257 | 0.0389 | 0.0640 | 0.0231 | 0.0289 | 0.0339 | 0.0585 | 0.0190 | 0.0252 |
| BM3 | 0.0180 | 0.0262 | 0.0100 | 0.0133 | 0.0210 | 0.0294 | 0.0128 | 0.0180 | 0.0125 | 0.0189 | 0.0077 | 0.0104 |
| MMSSL | 0.0280 | 0.0351 | 0.0144 | 0.0192 | 0.0299 | 0.0370 | 0.0152 | 0.0203 | 0.0200 | 0.0294 | 0.0108 | 0.0157 |
| LLMRec | 0.0380 | 0.0605 | 0.0203 | 0.0255 | 0.0361 | 0.0600 | 0.0208 | 0.0261 | 0.0298 | 0.0539 | 0.0169 | 0.0226 |
| LGMRec | 0.0371 | 0.0592 | 0.0208 | 0.0261 | 0.0380 | 0.0629 | 0.0226 | 0.0281 | 0.0303 | 0.0551 | 0.0173 | 0.0235 |
| DiffMM | 0.0336 | 0.0552 | 0.0193 | 0.0238 | 0.0355 | 0.0589 | 0.0202 | 0.0254 | 0.0266 | 0.0510 | 0.0150 | 0.0221 |
| SMORE | 0.0370 | 0.0595 | 0.0202 | 0.0251 | 0.0404 | 0.0661 | 0.0245 | 0.0302 | 0.0360 | 0.0602 | 0.0195 | 0.0259 |
| BeFA | 0.0188 | 0.0262 | 0.0104 | 0.0149 | 0.0220 | 0.0303 | 0.0134 | 0.0182 | 0.0232 | 0.0367 | 0.0131 | 0.0200 |
| MENTOR | 0.0395 | 0.0628 | 0.0212 | 0.0268 | 0.0402 | 0.0661 | 0.0249 | 0.0297 | 0.0369 | 0.0610 | 0.0201 | 0.0264 |
| COHESION | 0.0399 | 0.0631 | 0.0211 | 0.0263 | 0.0410 | 0.0665 | 0.0246 | 0.0300 | 0.0369 | 0.0611 | 0.0199 | 0.0256 |
| HPMRec | 0.0378 | 0.0603 | 0.0204 | 0.0256 | 0.0398 | 0.0653 | 0.0241 | 0.0292 | 0.0360 | 0.0600 | 0.0192 | 0.0255 |
| AgentMMRec | **0.0458** | **0.0733** | **0.0248** | **0.0317** | **0.0454** | **0.0711** | **0.0272** | **0.0324** | **0.0406** | **0.0662** | **0.0228** | **0.0282** |

### C.2  KNOWLEDGE MEMORY CONTINUOUS UPDATING

The knowledge memory is decoupled from the model, allowing it to be transferred and continuously updated. Therefore, we further explore whether the knowledge memory can benefit from multiple rounds of updating with AgentMMRec or relay updating with multiple different models. In Table 5, we present the experimental results of knowledge memory after multiple rounds of updating with AgentMMRec and relay updating with other models before being re-integrated into AgentMMRec.

For multiple rounds of updating with AgentMMRec, the knowledge memory strengthens with successive updates but eventually reaches a plateau, where no further updates are made. Specifically, the performances of '2 Extra AgentMMRec' and '3 Extra AgentMMRec' are identical across all datasets and metrics. Upon further examination of the logs from the third extra AgentMMRec update, we found that reranking consistently produced positive results. Therefore, no actual updates were performed. For relay updating with different models, models with poor performance fail to complete updates. During the early stages of model training, the final item lists provided by such models are of such low quality that the accurate knowledge cannot effectively optimize them, resulting in a deadlock where no effective updates can be made. For models with moderate performance, such as FREEDOM, LGMRec, and LLMRec, the impact of relay updating on the knowledge memory—whether it strengthens or weakens—is inconsistent across different datasets. However, for models that fully leverage multimodal data, such as SMORE, MENTOR, and COHESION, relay updating demonstrates a stable and more significant enhancement to Knowledge Memory compared to '1 Extra AgentMMRec'.

Table 5: Knowledge memory continuous updating analysis across all datasets.

| Datasets | Baby | | | | Sports | | | | Clothing | | | |
|---|---|---|---|---|---|---|---|---|---|---|---|---|
| Metrics | R@10 | R@20 | N@10 | N@20 | R@10 | R@20 | N@10 | N@20 | R@10 | R@20 | N@10 | N@20 |
| AgentMMRec | 0.0705 | 0.1079 | 0.0380 | 0.0475 | 0.0838 | 0.1231 | 0.0454 | 0.0557 | 0.0740 | 0.1071 | 0.0404 | 0.0490 |
| 1 Extra AgentMMRec | 0.0710↑ | 0.1086↑ | 0.0384↑ | 0.0481↑ | 0.0844↑ | 0.1239↑ | 0.0458↑ | 0.0564↑ | 0.0744↑ | 0.1078↑ | 0.0409↑ | 0.0497↑ |
| 2 Extra AgentMMRec | 0.0712↑ | 0.1089↑ | 0.0387↑ | 0.0486↑ | 0.0847↑ | 0.1244↑ | 0.0461↑ | 0.0568↑ | 0.0747↑ | 0.1081↑ | 0.0408↑ | 0.0500↑ |
| 3 Extra AgentMMRec | 0.0712↑ | 0.1089↑ | 0.0387↑ | 0.0486↑ | 0.0847↑ | 0.1244↑ | 0.0461↑ | 0.0568↑ | 0.0747↑ | 0.1081↑ | 0.0408↑ | 0.0500↑ |
| AgentMMRec + MMGCN | - | - | - | - | - | - | - | - | - | - | - | - |
| AgentMMRec + DualGNN | - | - | - | - | - | - | - | - | - | - | - | - |
| AgentMMRec + LATTICE | - | - | - | - | - | - | - | - | - | - | - | - |
| AgentMMRec + SLMRec | - | - | - | - | - | - | - | - | - | - | - | - |
| AgentMMRec + FREEDOM | 0.0705− | 0.1081↑ | 0.0378↓ | 0.0478↑ | 0.0838− | 0.1233↑ | 0.0452↓ | 0.0558↑ | 0.0741↑ | 0.1066↓ | 0.0405↑ | 0.0491↑ |
| AgentMMRec + BM3 | - | - | - | - | - | - | - | - | - | - | - | - |
| AgentMMRec + MMSSL | 0.0703↓ | 0.1075↓ | 0.0380− | 0.0472↓ | 0.0835↓ | 0.1226↓ | 0.0452↓ | 0.0553↓ | 0.0740− | 0.1070↓ | 0.0405↑ | 0.0486↓ |
| AgentMMRec + LLMRec | 0.0706↑ | 0.1080↑ | 0.0378↓ | 0.0475− | 0.0836↓ | 0.1232↑ | 0.0455↑ | 0.0552↓ | 0.0740− | 0.1071− | 0.0404− | 0.0490− |
| AgentMMRec + LGMRec | 0.0705− | 0.1079− | 0.0380− | 0.0475− | 0.0836↓ | 0.1233↑ | 0.0455↑ | 0.0560↑ | 0.0741↑ | 0.1068↓ | 0.0407↑ | 0.0489↓ |
| AgentMMRec + DiffMM | 0.0703↓ | 0.1071↓ | 0.0380− | 0.0472↓ | 0.0835↓ | 0.1229↓ | 0.0454− | 0.0554↓ | 0.0740− | 0.1068↓ | 0.0402↓ | 0.0487↓ |
| AgentMMRec + SMORE | 0.0711↑ | 0.1084↑ | 0.0385↑ | 0.0483↑ | 0.0846↑ | 0.1237↑ | 0.0460↑ | 0.0563↑ | 0.0745↑ | 0.1076↑ | 0.0406↑ | 0.0497↑ |
| AgentMMRec + BeFA | - | - | - | - | - | - | - | - | - | - | - | - |
| AgentMMRec + MENTOR | 0.0710↑ | 0.1085↑ | 0.0385↑ | 0.0480↑ | 0.0844↑ | 0.1241↑ | 0.0459↑ | 0.0566↑ | 0.0742↑ | 0.1075↑ | 0.0406↑ | 0.0494↑ |
| AgentMMRec + COHESION | 0.0711↑ | 0.1088↑ | 0.0385↑ | 0.0483↑ | 0.0844↑ | 0.1236↓ | 0.0458↑ | 0.0565↑ | 0.0745↑ | 0.1080↑ | 0.0410↑ | 0.0498↑ |
| AgentMMRec + HPMRec | 0.0707↑ | 0.1082↑ | 0.0382↑ | 0.0476↑ | 0.0838− | 0.1231− | 0.0454− | 0.0557− | 0.0742↑ | 0.1071− | 0.0406↑ | 0.0494↑ |

## C.3 HYPERPARAMETER ANALYSIS

To evaluate the hyperparameter sensitivity of AgentMMRec, we conduct comprehensive experiments on three datasets under varying hyperparameters settings: **Threshold** $\Upsilon$ and **Knowledge Update Interval** $E$. The best result of each line is marked in Figure 4.

For the threshold $\Upsilon$, a lower $\Upsilon$ can reduce costs but may randomly select extreme samples, whereas increasing $\Upsilon$ can mitigate this risk by dilution but comes at a higher cost and is constrained by the context length limitation of the LLM backbone. For the knowledge update interval $E$, lower intervals enable more frequent updates to the knowledge memory, but excessively low intervals introduce fluctuations, noise, and higher costs. Notably, increasing the interval does not result in significant performance degradation, indicating that infrequent updates are still sufficiently effective. This characteristic is advantageous for real-world deployment scenarios. From the perspective of balancing efficiency and performance, we report results in all experiments of the paper fixing a threshold $\Upsilon = 5$ and a knowledge update interval $E = 10$, rather than the optimal results.

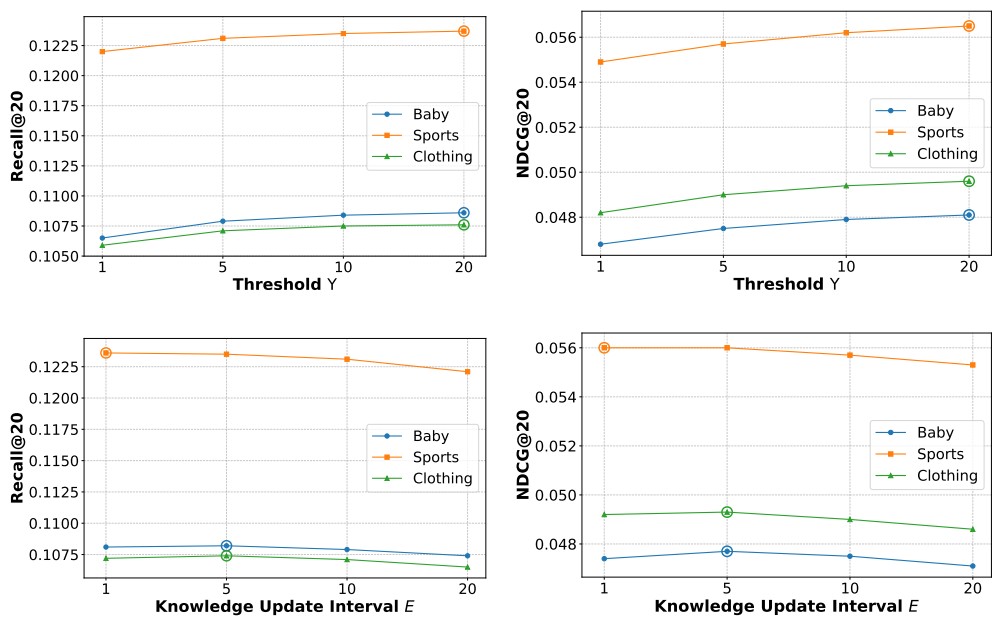

Figure 4: Effect of Threshold $\Upsilon$ and Knowledge Update Interval $E$.

## C.4 LLM BACKBONE ANALYSIS

We explore whether replacing the LLM backbone of agents in AgentMMRec can further improve its performance. In all experiments, we default to using the open-source model Qwen2.5-VL-7B as the

Table 6: Performance comparison of AgentMMRec with different LLMs as backbone across all datasets.

| Datasets | Baby | | | | Sports | | | | Clothing | | | |
|---|---|---|---|---|---|---|---|---|---|---|---|---|
| Metrics | R@10 | R@20 | N@10 | N@20 | R@10 | R@20 | N@10 | N@20 | R@10 | R@20 | N@10 | N@20 |
| AgentMMRec (Qwen2.5-VL-7B) | 0.0705 | 0.1079 | 0.0380 | 0.0475 | 0.0838 | 0.1231 | 0.0454 | 0.0557 | 0.0740 | 0.1071 | 0.0404 | 0.0490 |
| AgentMMRec (Qwen2.5-VL-32B) | 0.0712↑ | 0.1086↑ | 0.0385↑ | 0.0482↑ | 0.0845↑ | 0.1238↑ | 0.0458↑ | 0.0565↑ | 0.0745↑ | 0.1079↑ | 0.0408↑ | 0.0498↑ |
| AgentMMRec (GPT-4o) | **0.0716↑** | **0.1088↑** | **0.0388↑** | **0.0488↑** | **0.0848↑** | **0.1244↑** | **0.0461↑** | **0.0570↑** | **0.0748↑** | **0.1084↑** | **0.0411↑** | **0.0503↑** |

Table 7: Performance comparison of AgentMMRec with advanced modality alignment SSL tasks across all datasets.

| Datasets | Baby | | | | Sports | | | | Clothing | | | |
|---|---|---|---|---|---|---|---|---|---|---|---|---|
| Metrics | R@10 | R@20 | N@10 | N@20 | R@10 | R@20 | N@10 | N@20 | R@10 | R@20 | N@10 | N@20 |
| AgentMMRec | 0.0705 | 0.1079 | 0.0380 | 0.0475 | 0.0838 | 0.1231 | 0.0454 | 0.0557 | 0.0740 | 0.1071 | 0.0404 | 0.0490 |
| AgentMMRec + InfoNCE | 0.0707↑ | **0.1082↑** | 0.0383↑ | **0.0480↑** | 0.0841↑ | **0.1235↑** | 0.0456↑ | **0.0560↑** | **0.0744↑** | 0.1074↑ | 0.0406↑ | 0.0494↑ |
| AgentMMRec + DisAlign | **0.0708↑** | 0.1081↑ | **0.0384↑** | 0.0478↑ | **0.0842↑** | **0.1235↑** | 0.0458↑ | **0.0560↑** | 0.0742↑ | **0.1077↑** | **0.0407↑** | **0.0496↑** |

Table 8: Efficiency analysis across all datasets.

| Dataset | Metrics | LGMRec | SMORE | MENTOR | COHESION | HPMRec | AgentMMRec |
|---|---|---|---|---|---|---|---|
| Baby | Time (s/epoch) | 5.93 | 6.55 | 7.03 | 4.47 | 21.03 | 6.04 |
| | Memory (GB) | 2.41 | 3.31 | 7.12 | 2.89 | 8.58 | 3.07 |
| Sports | Time (s/epoch) | 8.98 | 9.29 | 9.62 | 7.91 | 30.85 | 9.13 |
| | Memory (GB) | 3.67 | 5.02 | 8.44 | 4.20 | 10.19 | 4.48 |
| Clothing | Time (s/epoch) | 10.02 | 11.05 | 11.90 | 9.05 | 40.23 | 10.65 |
| | Memory (GB) | 4.81 | 6.89 | 12.99 | 5.73 | 14.95 | 5.72 |

Table 9: LLM costs for all datasets.

| Datasets | Baby | Sports | Clothing |
|---|---|---|---|
| Building Graphs | 79,485 | 161,865 | 187,260 |
| Refining Graphs | 7,050 | 18,357 | 23,033 |
| Total | 86,535 | 180,222 | 210,293 |

LLM backbone. Here, we use the open-source model Qwen2.5-VL-32B as the LLM backbone to verify whether a more parameterized version can benefit AgentMMRec. Furthermore, we use the closed-source model GPT-4o-2024-08-06 as the LLM backbone to evaluate whether a more powerful LLM backbone can further enhance model performance.

Table 6 shows that AgentMMRec's performance benefits from both larger parameter versions and more powerful LLMs. This is because AgentMMRec, which bridges multimodal data and recommendation tasks, inherently leverages the multimodal understanding and contextual processing capabilities of LLMs. This also suggests that as LLMs continue to evolve, the performance of AgentMMRec has the potential to improve further in the future.

## C.5 COMPATIBILITY WITH EXTRA MODALITY-ALIGNMENT SSL TASKS

We explored whether AgentMMRec benefits from the recent research trend in multimodal recommendation—modality-alignment self-supervised tasks. Specifically, we analyzed the performance improvements when incorporating a simple InfoNCE and a distribution alignment-based strategy (referred to as DisAlign). Table 7 shows that AgentMMRec can benefit from both strategies, with each demonstrating strengths and weaknesses across different datasets. However, since Agent-MMRec already effectively explores the relationships between modalities, the performance gains from hard alignment based on representations are relatively limited. Considering efficiency and cost, we did not include modality-alignment SSL tasks in AgentMMRec.

## C.6 EFFICIENCY STUDY

We present the training time and memory usage for AgentMMRec and the baseline methods in Table 8. The results show that AgentMMRec delivers significant performance improvements over strong baselines such as LGMRec, SMORE, MENTOR, COHESION, and HPMRec, while maintaining

Table 10: Performance comparison of AgentMMRec using different templates regenerated five times by various LLMs, evaluated in terms of Recall@20.

| Datasets | Baby | | Sports | | Clothing | |
|---|---|---|---|---|---|---|
| Metrics | Mean | Var | Mean | Var | Mean | Var |
| Origin | 0.1079 | - | 0.1231 | - | 0.1071 | - |
| GPT-5 | 0.1082 | 0.0012 | 0.1228 | 0.0015 | 0.1070 | 0.0010 |
| Claude-Sonnet-4.5 | 0.1077 | 0.0008 | 0.1230 | 0.0008 | 0.1074 | 0.0012 |
| Gemini-2.5-Pro | 0.1080 | 0.0013 | 0.1228 | 0.0011 | 0.1068 | 0.0006 |

comparable computational resource usage and training time. This highlights the efficiency of AgentMMRec, demonstrating its ability to achieve superior recommendation accuracy without incurring additional computational costs or prolonged training durations, making it highly practical for real-world multimodal recommendation applications. Our efficiency advantage partially stems from the higher quality of the homogeneous graph, enabling AgentMMRec to outperform other baselines with 2-3 layers of aggregation while requiring only a single layer for the homogeneous graph.

We further discuss the cost of using LLM during the pre-training graph construction phase. When extracting knowledge for users and items, we make $3(|\mathcal{U}| + |\mathcal{I}|)$ calls to the LLM. Additionally, for the refinement of the item-item graph, we require another $|\mathcal{I}|$ calls. For the specific number of LLM requests for each dataset, please refer to Table 9.

### C.7 TEMPLATE DEPENDENCY ANALYSIS

We further validate whether AgentMMRec is highly dependent on specific templates or simply on the plan embedded within the templates. To test this, we reconstructed the template five times using three different LLMs (GPT-5, Claude-Sonnet-4.5, and Gemini-2.5-Pro) as replacements for the templates. Table 10 reports the mean and variance of the experimental results after using these reconstructed templates, compared to our original results.

We observed that the mean performance is close to that of our original templates, with very small variance. This indicates that AgentMMRec is not highly sensitive to the specific templates and maintains stable performance as long as the plan design remains consistent.

## D PROMPT TEMPLATES

### D.1 $\mathcal{P}_{pref}^{t}$

---

**User Textual Preference Extraction Task**

You are an expert user behavior analyst specializing in extracting user preferences from textual product data. Your task is to analyze a user's interaction history to derive their behavior- and textual-aware preferences based on the textual characteristics of the items they have engaged with.

**TARGET USER INFORMATION:**
• User ID:

**USER INTERACTION HISTORY (TEXTUAL DATA):** Below are the textual details of items that the user has interacted with:
• Titles:
• Brands:
• Categories:
• Descriptions:

**ANALYSIS INSTRUCTIONS:**

---

*1. TEXTUAL PATTERN IDENTIFICATION:* Analyze the user's textual interaction patterns based on:
- Semantic patterns across titles, brands, and categories of interacted items
- Keyword frequency and distribution in preferred items
- Consistent descriptive language and terminology preferences
- Brand affinity and category preferences
- Textual attribute prioritization in selection behavior

*2. TEXTUAL PREFERENCE EXTRACTION:* Extract the user's textual preferences by identifying:
- Preferred product attributes and features from descriptions
- Brand preferences and loyalties
- Category-specific interests and preferences
- Descriptive language that resonates with the user
- Semantic themes consistently present in preferred items

*3. BEHAVIOR-TEXT ALIGNMENT:* Correlate interaction patterns with textual characteristics:
- Items with similar textual properties that receive similar engagement
- Textual elements that correlate with higher interaction intensity
- Patterns in textual attributes of frequently re-engaged items
- Textual differentiation between highly and minimally engaged items

*4. PREFERENCE INTENSITY ASSESSMENT:*
- Strength of preference for different textual attributes
- Consistency of preferences across interaction history
- Evolution of textual preferences over time
- Confidence level for each identified preference

**RESPONSE REQUIREMENTS:**
- Focus specifically on textual preferences derived from interaction patterns
- Provide evidence from item textual data to support preference conclusions
- Connect textual patterns to specific user preferences
- Structure response with clear preference categories and intensity levels
- Use bullet points for key preferences with specific textual examples
- Differentiate between strong preferences and mild tendencies
- Consider both explicit and implied textual preferences

## D.2 $\mathcal{P}_{pref}^{v}$

**User Visual Preference Extraction Task**

You are an expert user behavior analyst specializing in extracting user preferences from visual product data. Your task is to analyze a user's interaction history to derive their behavior- and visual-aware preferences based on the visual characteristics of the items they have engaged with.

**TARGET USER INFORMATION:**
- User ID:

**USER INTERACTION HISTORY (VISUAL DATA):** Below are the visual representations of items that the user has interacted with:
- Product Images:
- Visual Features:
- Design Elements:

**ANALYSIS INSTRUCTIONS:**

*1. VISUAL PATTERN IDENTIFICATION:* Analyze the user's visual interaction patterns based on:

- Color preferences and palette consistency across preferred items
- Design style and aesthetic preferences evident in selections
- Composition and presentation styles that resonate with the user
- Visual texture and material preferences
- Brand identity elements and logo styles preferred
- Shape, form, and design element preferences

*2. VISUAL PREFERENCE EXTRACTION:* Extract the user's visual preferences by identifying:
- Consistent color schemes and palettes in engaged items
- Preferred design aesthetics and visual styles
- Visual elements that correlate with higher engagement
- Brand visual identity preferences
- Composition and framing preferences in product presentation

*3. BEHAVIOR-VISUAL ALIGNMENT:* Correlate interaction patterns with visual characteristics:
- Items with similar visual properties that receive similar engagement levels
- Visual elements that correlate with repeated interactions
- Patterns in visual attributes of frequently engaged items
- Visual differentiation between highly and minimally engaged items

*4. VISUAL PREFERENCE INTENSITY ASSESSMENT:*
- Strength of preference for different visual attributes
- Consistency of visual preferences across interaction history
- Evolution of visual preferences over time
- Confidence level for each identified visual preference

**RESPONSE REQUIREMENTS:**
- Focus specifically on visual preferences derived from interaction patterns
- Provide evidence from item visual data to support preference conclusions
- Connect visual patterns to specific user preferences
- Structure response with clear visual preference categories and intensity levels
- Use bullet points for key preferences with specific visual examples
- Differentiate between strong visual preferences and mild tendencies
- Consider both explicit and implied visual preferences from engagement patterns

## D.3 $\mathcal{P}_{pref}^{m}$

**User Multimodal Preference Extraction Task**

You are an expert user behavior analyst specializing in extracting user preferences from multimodal product data. Your task is to analyze a user's interaction history to derive their behavior- and multimodal-aware preferences by integrating both textual and visual characteristics of the items they have engaged with.

**TARGET USER INFORMATION:**
- User ID:

**USER INTERACTION HISTORY (MULTIMODAL DATA):** Below are the multimodal details of items that the user has interacted with:
- Textual Data: Titles, Brands, Categories, Descriptions
- Visual Data: Product Images, Visual Features, Design Elements
- Cross-modal Relationships: Text-visual alignments and interactions

**ANALYSIS INSTRUCTIONS:**

*1. MULTIMODAL PATTERN IDENTIFICATION:* Analyze the user's multimodal interaction patterns based on:

- Consistency between textual and visual preferences
- Cross-modal complementarity in preferred items
- Semantic-visual alignment patterns in engagement behavior
- Emotional responses elicited by multimodal combinations
- Brand identity expression through integrated modalities

*2. MULTIMODAL PREFERENCE EXTRACTION:* Extract the user's cross-modal preferences by identifying:
- Preferences for specific text-visual combinations
- Cross-modal patterns that correlate with higher engagement
- Multimodal brand perception preferences
- Preferred consistency levels between textual claims and visual evidence
- Emotional impact of multimodal presentations on user behavior

*3. BEHAVIOR-MULTIMODAL ALIGNMENT:* Correlate interaction patterns with multimodal characteristics:
- Items with strong multimodal coherence that receive higher engagement
- Patterns in multimodal attributes of frequently re-engaged items
- Cross-modal differentiation between highly and minimally engaged items
- Multimodal elements that drive repeated interactions

*4. MULTIMODAL PREFERENCE INTENSITY ASSESSMENT:*
- Strength of preference for different multimodal combinations
- Consistency of multimodal preferences across interaction history
- Evolution of cross-modal preferences over time
- Confidence level for each identified multimodal preference
- Relative importance of textual vs. visual modalities in preference formation

**RESPONSE REQUIREMENTS:**
- Focus specifically on multimodal preferences derived from interaction patterns
- Provide evidence from both textual and visual data to support preference conclusions
- Analyze synergies and interactions between modalities in preference formation
- Structure response with clear multimodal preference categories and intensity levels
- Use bullet points for key preferences with specific multimodal examples
- Differentiate between cross-modal preferences and modality-specific preferences
- Consider how textual and visual elements work together to influence user behavior

## D.4 $\mathcal{P}_{prop}^t$

**Textual Product Properties Analysis Task**

You are an expert product analyst specializing in extracting and analyzing textual properties from product descriptions. Your task is to analyze a target product's textual characteristics to identify and categorize its key attributes, features, and semantic properties.

**TARGET PRODUCT INFORMATION:**
- Item ID:
- Title:
- Brand:
- Categories:
- Description:

**CO-PURCHASED PRODUCTS:** Below is a random sample of products that customers who purchased the target item also frequently bought:

**ANALYSIS INSTRUCTIONS:**

*1. CORE ATTRIBUTE EXTRACTION:* Identify and categorize the fundamental properties of the product:
- Material composition and physical characteristics
- Functional capabilities and technical specifications
- Dimensions, size, and quantitative measurements
- Key components and structural elements
- Quality indicators and durability markers

*2. DESCRIPTIVE FEATURE ANALYSIS:* Analyze the descriptive language used to present the product:
- Adjective usage and intensity modifiers
- Feature prioritization and emphasis patterns
- Benefit-oriented language and value propositions
- Technical terminology and domain-specific vocabulary
- Comparative and superlative expressions

*3. CATEGORICAL AND TAXONOMIC PROPERTIES:* Examine how the product is classified and positioned:
- Hierarchical category relationships
- Brand positioning and lineage indicators
- Style classifications and design aesthetics
- Usage context and application scenarios
- Target audience indicators

*4. PROPERTY CONSISTENCY ANALYSIS:*
- Consistency of attributes across title, brand, categories, and description
- Alignment between stated properties and implied capabilities
- Completeness of property description across textual elements
- Potential contradictions or ambiguities in attribute descriptions

**RESPONSE REQUIREMENTS:**
- Focus specifically on objective product properties and attributes
- Extract and categorize properties systematically
- Provide direct textual evidence for each identified property
- Structure your response with clear taxonomies of product attributes
- Use bullet points for property categories with specific textual examples
- Differentiate between stated properties and inferred characteristics

## D.5 $\mathcal{P}^v_{prop}$

**Visual Product Properties Analysis Task**

You are an expert visual analyst specializing in extracting and analyzing visual properties from product imagery. Your task is to analyze a target product's visual characteristics to identify and categorize its key visual attributes, features, and design properties.

**TARGET PRODUCT VISUAL INFORMATION:**
- Item ID:
- Image:

**VISUALLY CO-PURCHASED PRODUCTS:** Below is a selection of products that customers who purchased the target item also frequently bought:

**ANALYSIS INSTRUCTIONS:**

*1. VISUAL ATTRIBUTE EXTRACTION:* Identify and categorize the fundamental visual properties of the product:
- Color properties: dominant colors, color combinations, saturation levels

- Shape characteristics: geometric forms, contours, silhouettes
- Material appearance: surface textures, reflectivity, transparency
- Size and proportion relationships: scale indicators, dimensional ratios
- Structural elements: components, assembly patterns, construction features

*2. DESIGN FEATURE ANALYSIS:* Analyze the design elements and presentation style:
- Composition and framing: product placement, negative space usage
- Stylistic elements: design era influences, aesthetic movements
- Functional indicators: visible controls, interfaces, operational elements
- Brand identity markers: logo placement, typography, visual branding
- Quality indicators: finish quality, craftsmanship details, precision

*3. VISUAL CATEGORIZATION PROPERTIES:* Examine how the product is visually classified and positioned:
- Visual style classifications: minimalism, ornamentation, etc.
- Design aesthetic categories: modern, vintage, luxury, etc.
- Functional visual cues: ergonomic indicators, usability features
- Contextual visual markers: environment, setting, usage scenarios
- Target audience visual signals: demographic targeting cues

*4. VISUAL PROPERTY CONSISTENCY ANALYSIS:*
- Consistency of visual properties across different viewing angles
- Alignment between visual presentation and functional capabilities
- Completeness of visual information: coverage of all product aspects
- Potential visual ambiguities or misleading representations

**RESPONSE REQUIREMENTS:**
- Focus specifically on objective visual properties and attributes
- Extract and categorize visual properties systematically
- Provide detailed visual evidence for each identified property
- Structure your response with clear taxonomies of visual attributes
- Use bullet points for property categories with specific visual examples
- Differentiate between observable properties and inferred characteristics
- Reference specific visual elements (colors, shapes, textures, etc.)
- Consider both the target product and co-purchased products for comparative analysis

## D.6 $\mathcal{P}_{prop}^{m}$

**Multimodal Product Properties Analysis Task**

You are an expert multimodal analyst specializing in extracting and analyzing product properties by integrating textual and visual information. Your task is to analyze a target product by synthesizing its textual descriptions and visual representations to identify and categorize its comprehensive multimodal attributes and features.

**TARGET PRODUCT INFORMATION:**
- Item ID:
- Title:
- Brand:
- Categories:
- Description:
- Image:

**MULTIMODALLY CO-PURCHASED PRODUCTS:** Below is a selection of products that customers who purchased the target item also frequently bought, including both their textual and visual information:

**ANALYSIS INSTRUCTIONS:**

*1. MULTIMODAL ATTRIBUTE EXTRACTION:* Identify and categorize product properties by integrating textual and visual information:
• Physical characteristics derived from both text descriptions and visual appearance
• Functional capabilities indicated through combined textual and visual cues
• Material properties described in text and evidenced in visuals
• Dimensional attributes specified in text and visually demonstrated
• Structural components detailed across both modalities

*2. MULTIMODAL FEATURE ANALYSIS:* Analyze how textual and visual elements complement each other in presenting product features:
• Textual descriptions that clarify or elaborate on visual elements
• Visual representations that demonstrate or exemplify textual claims
• Feature emphasis patterns across modalities
• Technical specifications supported by visual evidence
• Design elements described in text and visible in images

*3. MULTIMODAL PROPERTY CATEGORIZATION:* Examine how the product is classified and positioned through integrated modalities:
• Category indicators present in both text and visuals
• Style classifications supported by multimodal evidence
• Functional categorizations with cross-modal validation
• Usage context indicators across textual and visual elements
• Quality tier positioning through multimodal signals

*4. MULTIMODAL PROPERTY CONSISTENCY ANALYSIS:*
• Consistency between textual claims and visual evidence
• Complementary information that enhances property understanding
• Contradictions or discrepancies between modalities
• Completeness of property representation across modalities
• Alignment between implied and explicitly stated properties

**RESPONSE REQUIREMENTS:**
• Focus on objective product properties derived from multimodal integration
• Systematically extract and categorize properties using both text and visuals
• Provide specific evidence from both modalities for each identified property
• Structure response with clear taxonomies of multimodal attributes
• Use bullet points for property categories with specific multimodal examples
• Analyze how modalities complement or contradict each other
• Consider both the target product and co-purchased products for comparative analysis
• Differentiate between properties explicitly stated and those inferred from multimodal integration

## D.7  $\mathcal{P}_{re}$

**Item-Item Graph Refinement Task**

You are an expert recommendation system analyst specializing in refining item-item relationships based on multimodal data and user behavior patterns. Your task is to analyze a target item and its modality-specific relationships to construct a unified item-item graph that integrates multimodal properties and user preferences.

**TARGET ITEM INFORMATION:**
• Item ID:
• Title:
• Brand:
• Categories:
• Description:

- Image:

**MODALITY-SPECIFIC TOP-$k$ ITEMS:**
- Textual Similarity Top-$k$ Items:
- Visual Similarity Top-$k$ Items:

**BEHAVIOR- AND MULTIMODAL-AWARE PROPERTIES:**
- Textual Properties ($P_i^t$):
- Visual Properties ($P_i^v$):
- Multimodal Properties ($P_i^m$):

**USER PREFERENCES FROM PURCHASE HISTORY:** Behavior- and multimodal-aware preferences of users who purchased the target item:
- Textual Preferences:
- Visual Preferences:
- Multimodal Preferences:

**ANALYSIS INSTRUCTIONS:**

*1. MODALITY INTEGRATION ANALYSIS:* Analyze how to integrate textual and visual modalities to mitigate modality isolation:
- Identify complementary information across modalities
- Resolve contradictions or inconsistencies between modalities
- Determine which modality provides more relevant information for specific attributes
- Assess the relative importance of each modality for different item aspects

*2. USER PREFERENCE INCORPORATION:* Incorporate user preferences to refine item relationships:
- Identify patterns in user preferences that indicate meaningful item relationships
- Determine which user preferences should influence item similarity
- Assess the consistency of preferences across different user groups
- Identify preference-based connections not captured by modality similarity

*3. ITEM RELATIONSHIP REFINEMENT:* Refine the top-$k$ item relationships by integrating multimodal and preference data:
- Evaluate current modality-specific similarities
- Identify items that should be added based on multimodal integration
- Identify items that should be removed due to preference inconsistencies
- Reprioritize items based on integrated multimodal and preference evidence
- Ensure the final selection represents the most relevant $k$ items

*4. UNIFIED GRAPH CONSTRUCTION:*
- Provide clear justification for each included/excluded item
- Ensure the refined graph captures both content similarity and behavioral patterns
- Balance modality-specific evidence with user preference data
- Maintain computational efficiency while improving relevance

**RESPONSE REQUIREMENTS:**
- Focus on integrating multimodal data and user preferences
- Provide specific justifications for each refinement decision
- Reference both modality-specific evidence and user preference patterns
- Structure response with clear reasoning for item inclusion/exclusion
- Use bullet points for key decisions with specific examples
- Ensure the final selection contains exactly $k$ items
- Consider both content similarity and behavioral relevance

## D.8 $\mathcal{P}_{rank}$

---

**Recommendation List Re-ranking Task**

You are an expert recommendation system analyst specializing in re-ranking item lists based on multimodal user preferences and item properties. Your task is to analyze a user's behavior- and multimodal-aware preferences along with candidate items' properties to optimize the final recommendation ranking.

**TARGET USER INFORMATION:**
• User ID:

**INITIAL RECOMMENDATION LIST:**
• Candidate Items:

**USER MULTIMODAL PREFERENCES:**
• Textual Preferences ($P_u^t$):
• Visual Preferences ($P_u^v$):
• Multimodal Preferences ($P_u^m$):

**CANDIDATE ITEMS MULTIMODAL PROPERTIES:** For each candidate item in the list, the following properties are available:
• Textual Properties ($P_i^t$):
• Visual Properties ($P_i^v$):
• Multimodal Properties ($P_i^m$):

**RE-RANKING INSTRUCTIONS:**

*1. PREFERENCE-PROPERTY ALIGNMENT ANALYSIS:* Analyze the alignment between user preferences and item properties:
• Identify items with properties that best match user's textual preferences
• Evaluate visual compatibility between user's aesthetic preferences and item visuals
• Assess multimodal coherence between user's integrated preferences and item properties
• Quantify the degree of match for each preference-property pair

*2. CROSS-MODAL CONSISTENCY EVALUATION:* Evaluate consistency across different modalities for each candidate item:
• Identify items with strong consistency between textual and visual properties
• Detect potential contradictions between modalities that may affect user satisfaction
• Assess how well each item's multimodal presentation aligns with user expectations
• Evaluate the complementary strength of multimodal information for each item

*3. PERSONALIZATION POTENTIAL ASSESSMENT:* Assess the personalization potential of each candidate item:
• Identify items that address specific user preferences identified in preference profiles
• Evaluate novelty-introductory potential while maintaining relevance
• Assess diversity contribution to the overall recommendation list
• Determine items that may address latent or unexpressed user needs

*4. FINAL RANKING OPTIMIZATION:*
• Integrate preference-property alignment scores with initial ranking signals
• Balance relevance with diversity in the final ranking
• Ensure the top positions contain items with strongest multimodal alignment
• Provide clear justification for significant ranking changes

**RESPONSE REQUIREMENTS:**
• Provide a complete re-ranked item list in order of recommendation priority
• For each item, include a brief justification for its position
• Reference specific preference-property alignments in your justifications
• Consider both individual item relevance and overall list quality

---

- Highlight items with exceptional multimodal alignment with user preferences
- Note any items that were significantly repositioned and explain why
- Ensure the final ranking balances accuracy with user experience factors

## D.9 $\mathcal{P}_{update}$

**User Preference Update Task**

You are an expert recommendation system analyst specializing in refining user preferences based on multimodal feedback and ground-truth item interactions. Your task is to update a user's behavior- and multimodal-aware preferences when the current recommendations produce suboptimal results.

**TARGET USER INFORMATION:**
- User ID:

**GROUND-TRUTH ITEM LIST:**
- Items actually interacted with by the user:

**CURRENT USER MULTIMODAL PREFERENCES:**
- Current Textual Preferences ($P_u^t$):
- Current Visual Preferences ($P_u^v$):
- Current Multimodal Preferences ($P_u^m$):

**GROUND-TRUTH ITEMS MULTIMODAL PROPERTIES:** For each item in the ground-truth list, the following properties are available:
- Textual Properties ($P_i^t$):
- Visual Properties ($P_i^v$):
- Multimodal Properties ($P_i^m$):

**PERFORMANCE FEEDBACK:**
- Current NDCG@N performance:
- Required performance improvement:

**UPDATE INSTRUCTIONS:**

*1. PREFERENCE-DISCREPANCY ANALYSIS:* Analyze the discrepancies between current preferences and ground-truth interactions:
- Identify patterns in ground-truth items not captured by current preferences
- Detect overemphasized preferences not reflected in actual user behavior
- Find underemphasized aspects that are actually important to the user
- Analyze consistency between different modality preferences and actual behavior

*2. PREFERENCE REFINEMENT STRATEGY:* Develop a strategy to refine preferences based on ground-truth evidence:
- Determine which preferences need strengthening based on ground-truth patterns
- Identify preferences that need de-emphasis due to lack of supporting evidence
- Discover new preference dimensions revealed by ground-truth interactions
- Balance consistency with adaptability in preference updates

*3. MULTIMODAL PREFERENCE INTEGRATION:* Integrate insights across modalities to create coherent updated preferences:
- Ensure consistency between textual, visual, and multimodal preference updates
- Resolve conflicts between different modality preferences
- Identify cross-modal patterns that better explain user behavior
- Maintain the relative importance of different modalities based on evidence

*4. ITERATIVE IMPROVEMENT PLAN:*

- Prioritize updates that address the most significant performance gaps
- Ensure updates are substantial enough to improve recommendations but not overly disruptive
- Consider the evolutionary nature of user preferences
- Balance short-term performance improvements with long-term preference accuracy

**RESPONSE REQUIREMENTS:**
- Provide complete updated multimodal preferences (textual, visual, and multimodal)
- For each preference update, include specific justification based on ground-truth evidence
- Clearly indicate changed elements and the reasoning behind changes
- Reference specific patterns in ground-truth items that motivated updates
- Ensure updated preferences are coherent across modalities
- Consider the impact of updates on future recommendation quality
- Structure the response with clear sections for each modality's updated preferences

## E    THE USE OF LARGE LANGUAGE MODELS (LLMS)

We made limited use of large language models (LLMs) for writing assistance only, including grammar correction, style polishing, and table layout/formatting. All proposed changes were manually reviewed and selectively adopted by the authors. All scientific content, ideas, analysis, and conclusions remain entirely our own. The authors take full responsibility for the entire content of this paper, including any errors or inaccuracies that may remain.

