# OpenReview forum: "Agents as Knowledge Integrator and Utilizer in Multimodal Recommendation"
_ICLR.cc/2026/Conference — Submitted to ICLR 2026_

### Official Review · Reviewer_4vcD · 2025-10-27

**Soundness:** 3
**Presentation:** 3
**Contribution:** 3
**Rating:** 4
**Confidence:** 3

**Summary:**

AgentMMRec is an LLM-powered agent framework for multimodal recommendation that tries to close the semantic gap between raw vision/text signals and collaborative-filtering objectives. The whole loop is pre-computed or run every E epochs to keep cost acceptable.
On three Amazon subsets (Baby, Sports, Clothing) the system beats 15 baselines and shows good robustness under data sparsity and item-cold-start. Code is provided.

**Strengths:**

1. Strong reproducibility, containing full prompts, statistics, and implementation details are given; anonymous GitHub link provided.
2. Feeding the agent-constructed graphs or re-ranker into existing models (SMORE, MENTOR, etc.) improves them, showing the memory is model-agnostic.

**Weaknesses:**

1. Scalability: Context-length limit Υ=5 neighbours is tiny; authors admit Υ>20 is infeasible.
2. Only Amazon review data (text + one image) are used; no short-video, audio, or social-media modalities.
3. Data leakage risk – the same LLM that reads item text/images at test time also generates training graphs; it may trivially match items by brand/title strings, inflating cold-start numbers.

**Questions:**

1. What is the end-to-end training time versus the strongest baseline on the largest dataset?
2. What happens if you replace the 7B-VL model with a 0.5B distilled LM – is the framework still cost-effective?

---

> ### Author Response · Authors · 2025-11-19
>
> Thanks for your comments. We will address your questions point by point.
>
>
>
> **Weakness 1**
>
> > For a discussion on different values of $\Upsilon$, please refer to **Appendix C.3**. Thanks to the high reliability of interaction information and our knowledge update process, which further ensures the quality of knowledge based on reranking results, the experimental results reported throughout the paper are under the setting of $\Upsilon = 5$, demonstrating the effectiveness of this configuration. When using LLM backbones with stronger context understanding capabilities, the value of $\Upsilon$ can be further increased based on resources and requirements. This indicates that AgentMMRec can "evolve" alongside the rapid advancements in LLM technology.
>
> **Weakness 2**
>
> > In fact, in the current field of multimodal recommendation, most works only consider text and images (please refer to the latest survey [1] and its continuously updated awesome list repository). Therefore, we selected the most commonly used datasets to ensure fairness in comparison.
> >
> > It is worth mentioning that the current capabilities of MLLMs in handling audio and video are still not as advanced as their capabilities in processing visual data. However, as MLLMs continue to “evolve”, AgentMMRec can easily adapt to scenarios involving additional modalities in the future.
>
> **Weakness 3**
>
> > The homogeneous graph is not directly generated by the LLM. Instead, the LLM generates or extracts knowledge, and the homogeneous graph is constructed based on the representation similarity of the knowledge (see **Eq.10** and **Eq.11**). Additionally, **Eq.12** describes how the LLM performs selective aggregation on the multimodal homogeneous graph, which is still constructed based on representation similarity. Therefore, the concerns you raised will not occur.
> >
> > Furthermore, this homogeneous graph is beneficial for cold-start scenarios. In the absence of historical interactions, cold-start items rely heavily on multimodal information. Thus, this approach is advantageous. Please refer to the cold-start experimental results provided in **Appendix C.1**.
>
> **Question 1**
>
> > We have added the end-to-end training time of AgentMMRec under the standard settings on three datasets, compared to the strong baselines, in the appendix (highlighted in blue in **Appendix C.6**).
>
> **Question 2**
>
> > Qwen2.5 0.5B lacks multimodal understanding capabilities and can only process text, which is why we did not choose it as an LLM backbone. In contrast, Qwen2.5-VL-7B has both low deployment and inference costs and can be efficiently deployed on commercial-grade GPUs.
> >
> > Additionally, in **Appendix C.4**, we provide experimental results using larger open-source LLMs and more powerful closed-source LLMs (e.g., **Qwen2.5-VL-32B** and **GPT-4o**), further validating that AgentMMRec can benefit from stronger LLM backbones.
>
>
>
> Refs
>
> [1] A Survey on Multimodal Recommender Systems: Recent Advances and Future Directions, TMM 2025.
>
>
>
> We sincerely appreciate your valuable comments and hope our response has adequately addressed your concerns, providing sufficient reasons to raise the score. We are pleased to address any further concerns you may have. Wishing you all the best!
>
>
>
> > Friendly reminder: ICLR allows the submission of revised versions during the rebuttal period. Please refer directly to the revised version to review the manuscript updates.

---

> ### Author Response · Authors · 2025-11-26
> **Supplement of response to Reviewer 4vcD**
>
> For your convenience, we have provided the updated experimental results in the comments. You can also find all revised parts in the revised version.
>
> For **End-to-end training time**
>
> > We have added the end-to-end training time of AgentMMRec under the standard settings on three datasets, compared to the strong baselines, in the appendix (highlighted in blue in **Appendix C.6**).
>
> **Table:** Efficiency analysis across all datasets.
>
> | Dataset  | Metrics        | LGMRec | SMORE | MENTOR | COHESION | HPMRec | AgentMMRec |
> | -------- | -------------- | ------ | ----- | ------ | -------- | ------ | ---------- |
> | Baby     | Time (s/epoch) | 5.93   | 6.55  | 7.03   | 4.47     | 21.03  | 6.04       |
> |          | Memory (GB)    | 2.41   | 3.31  | 7.12   | 2.89     | 8.58   | 3.07       |
> | Sports   | Time (s/epoch) | 8.98   | 9.29  | 9.62   | 7.91     | 30.85  | 9.13       |
> |          | Memory (GB)    | 3.67   | 5.02  | 8.44   | 4.20     | 10.19  | 4.48       |
> | Clothing | Time (s/epoch) | 10.02  | 11.05 | 11.90  | 9.05     | 40.23  | 10.65      |
> |          | Memory (GB)    | 4.81   | 6.89  | 12.99  | 5.73     | 14.95  | 5.72       |
>
>
>
> This is a supplement to the above rebuttal. All updates can be found in the revised version.
>
> We sincerely appreciate your valuable comments and hope our response has adequately addressed your concerns, providing sufficient reasons to raise the score. We are pleased to address any further concerns you may have, and we wish you all the best!

---

### Official Review · Reviewer_HDYt · 2025-10-31

**Soundness:** 3
**Presentation:** 3
**Contribution:** 2
**Rating:** 4
**Confidence:** 3

**Summary:**

The paper proposes AgentMMRec, an agent-based multimodal recommendation framework leveraging LLMs to bridge the semantic gap between multimodal data and recommendation tasks. The architecture contains two agents — an Integrator Agent to infer user preferences and item properties from multimodal data and store them in a knowledge memory, and a Utilizer Agent to refine item-item graphs and rerank recommendations using this stored knowledge. Memory updates are triggered based on reranking performance feedback via prompt templates. Extensive experiments across multiple datasets show performance gains over multimodal graph-based baselines under normal and sparse data situations.

**Strengths:**

- Well-organized paper with thorough explanation of the proposed framework and clear methodology steps.
- Comprehensive experimental evaluation on multiple datasets, including sparsity and cold-start scenarios.
- The compatibility analysis showing benefits when components are transferred to other baselines is interesting and well reported.

**Weaknesses:**

- The graph structure improvement is incremental compared to existing multimodal + graph works such as; novelty is mainly in the systemization of components rather than in core graph construction. So the main innovation in my view is  "memory update via feedback". However, it is relatively simple and heavily reliant on prompt-template-guided LLM output, without deeper algorithmic or modeling advances. Therefore, the overall innovation point is still lacking.

- Heavy reliance on LLMs for preference/property extraction may limit reproducibility and interpretability, and much of the novelty lies in engineering details rather than theoretical contributions.

**Questions:**

- Could the authors clarify why the proposed memory update strategy, which is prompt-controlled, is preferable over more sophisticated memory storage research or methods?
- Is there any analysis of the robustness of the knowledge memory to noisy or biased prompt outputs from LLMs?

---

> ### Author Response · Authors · 2025-11-19
>
> Thanks for your comments. We will address your questions point by point.
>
>
>
> **Weakness 1**
>
> > We would like to clarify the novelty of AgentMMRec. Compared to existing LLM-based recommendation frameworks, AgentMMRec introduces the following unique innovations:
> >
> > 1. This novel design ensures that LLMs can fully utilize their multimodal understanding and contextual reasoning capabilities to better align multimodal information with recommendation tasks. Additionally, the preconstructed graphs based on raw features and historical interactions require minimal updates during training, significantly reducing the inherent high computational overhead of LLM-based recommender systems. This approach is both efficient and unique, distinguishing it from prior work.
> > 2. Unlike previous methods, our reranking mechanism is not only used to reorder the final recommendation list but also to enhance the quality of the maintained knowledge. While constructing homogeneous graphs, we simultaneously maintain a corresponding knowledge memory, which is utilized during the reranking stage. The evaluation from reranking is further used to refine and optimize the knowledge quality, forming a cyclic evolutionary process. This innovative design, integrating reranking with knowledge optimization, is a key feature absent in prior works.
> > 3. Unlike existing LLM-based recommendation models, the homogeneous graphs we construct can be directly integrated into existing multimodal recommendation models, resulting in significant performance improvements (see **Table 2 in Section 4.4**). Furthermore, the maintained knowledge memory can be seamlessly transferred to existing models and even jointly optimized across multiple models (see **Table 5 in Appendix C.2**). These designs and advantages are unique to our approach and not present in previous works.
>
> **Weakness 2**
>
> > Great question! In fact, AgentMMRec does not heavily rely on the specific templates but instead focuses more on the plan within the template, which provides the detailed task requirements. To address this, we conducted an interesting additional experiment (highlighted in blue in **Appendix C.7**). In this experiment, we regenerated the template five times using three different LLMs and replaced the original templates. We then re-ran the experiments on the all dataset and calculated the mean and variance of the results.
> >
> > We observed that the mean performance was close to that of the original templates, with very small variance. This indicates that AgentMMRec is not highly sensitive to the specific templates and maintains stable performance as long as the plan design remains consistent. For details on the innovations, please refer to our response to **Weakness 1**.
>
> **Question 1**
>
> > Please refer to **Section 3.4**, where the knowledge update process is triggered when reranking fails. Therefore, we choose to update the knowledge using the correct sequence and multimodal information through the LLM. The effectiveness of the knowledge update is then validated based on the results of the subsequent reranking. This design ensures the reliability and effectiveness of the knowledge updates. Additionally, the experiments on continuous updates of the knowledge memory in **Appendix C.2** demonstrate that performance saturates after multiple updates, indicating that AgentMMRec is not sensitive to occasional noise from the LLM.
>
> **Question 2**
>
> > We provide a discussion and experiments on the impact of template selection in our response to **Weakness 2**. In fact, since our knowledge is updated based on the reranking results and validated through subsequent reranking, this design greatly mitigates the noise and hallucination issues introduced by the LLM.
>
>
>
> We sincerely appreciate your valuable comments and hope our response has adequately addressed your concerns, providing sufficient reasons to raise the score. We are pleased to address any further concerns you may have. Wishing you all the best!
>
>
>
> > Friendly reminder: ICLR allows the submission of revised versions during the rebuttal period. Please refer directly to the revised version to review the manuscript updates.

---

> ### Author Response · Authors · 2025-11-26
> **Supplement of response to Reviewer HDYt**
>
> For your convenience, we have provided the updated experimental results in the comments. You can also find all revised parts in the revised version.
>
> For **Template dependencies**
>
> > AgentMMRec does not heavily rely on the specific templates but instead focuses more on the plan within the template, which provides the detailed task requirements. To address this, we conducted an interesting additional experiment (highlighted in blue in **Appendix C.7**). In this experiment, we regenerated the template five times using three different LLMs and replaced the original templates. We then re-ran the experiments on the all dataset and calculated the mean and variance of the results.
> >
> > We observed that the mean performance was close to that of the original templates, with very small variance. This indicates that AgentMMRec is not highly sensitive to the specific templates and maintains stable performance as long as the plan design remains consistent.
>
> **Table:** Performance comparison of AgentMMRec using different templates regenerated five times by various LLMs, evaluated in terms of Recall@20.
>
> | Datasets          | Baby   |        | Sports |        | Clothing |        |
> | ----------------- | ------ | ------ | ------ | ------ | -------- | ------ |
> | Metrics           | Mean   | Var    | Mean   | Var    | Mean     | Var    |
> | Origin            | 0.1079 | -      | 0.1231 | -      | 0.1071   | -      |
> | GPT-5             | 0.1082 | 0.0012 | 0.1228 | 0.0015 | 0.1070   | 0.0010 |
> | Claude-Sonnet-4.5 | 0.1077 | 0.0008 | 0.1230 | 0.0008 | 0.1074   | 0.0012 |
> | Gemini-2.5-Pro    | 0.1080 | 0.0013 | 0.1228 | 0.0011 | 0.1068   | 0.0006 |
>
>
>
> This is a supplement to the above rebuttal. All updates can be found in the revised version.
>
> We sincerely appreciate your valuable comments and hope our response has adequately addressed your concerns, providing sufficient reasons to raise the score. We are pleased to address any further concerns you may have, and we wish you all the best!

---

### Official Review · Reviewer_M43Y · 2025-11-01

**Soundness:** 3
**Presentation:** 3
**Contribution:** 3
**Rating:** 6
**Confidence:** 3

**Summary:**

This paper addresses the "semantic gap" in multimodal recommendation—the gap between raw data (images, text) and the actual recommendation task. The authors propose AgentMMRec, a novel framework where Large Language Models (LLMs) act as agents to bridge this gap. A feedback loop is included where re-ranking performance is evaluated, and if a re-ranking is suboptimal, the Integrator Agent updates the user's preferences in the knowledge memory. Experiments on three Amazon datasets show that AgentMMRec significantly outperforms a wide range of state-of-the-art multimodal recommendation baselines on standard ranking metrics.

**Strengths:**

- The (Integrator, Utilizer) agent framework is the paper's primary strength. It provides a clean and powerful new paradigm for using LLMs in recommendation, separating the intensive reasoning/knowledge-generation phase from the application/utilization phase.
- The paper's core premise is to replace noisy, raw multimodal features with "knowledge" (inferred preferences $P_u$ and properties $P_i$) that is explicitly generated by an LLM to be task-relevant. This is a more direct and intelligent approach than traditional feature-fusion or alignment methods.
- The "knowledge memory" is a powerful and modular concept. The compatibility analysis in Table 2, which shows that transferring the "knowledge" (via +Graph or +Rerank) boosts the performance of other SOTA models, is extremely strong evidence for the paper's claims. It proves that the generated knowledge is high-quality and generally useful.
- The framework achieves significant gains over a very strong and comprehensive set of recent baselines (including models from 2023-2025) on all datasets (Table 1).
- The method's ability to infer item properties $P_i$ directly from multimodal data makes it highly effective for item cold-start (Table 4), which is a major practical advantage.

**Weaknesses:**

- The framework, as-proposed, is enormously complex and computationally expensive. The "pre-building" phase requires multiple LLM calls (using a 7B VLM) for every single user and every single item to generate the initial knowledge memory and graphs (Eqs. 1-8, 12). Then, it requires more LLM calls during training (every $E$ epochs) for re-ranking and knowledge updates (Eqs. 13-14). The paper dismisses this as a "pre-building" cost, but for any real-world dataset with millions of users and items, this phase is computationally intractable. A thorough cost and latency analysis is completely missing.
- The system's performance is critically dependent on a large and complex set of nine different, hand-crafted prompt templates (Appendices D.1-D.9). This introduces a significant and fragile "hidden" hyperparameter-tuning problem. It is unclear how robust this prompt-based knowledge generation is or how well it would generalize to new domains without extensive re-engineering of the prompts.
- The paper uses a threshold $Y$ (e.g., $Y=5$ in Appendix C.3) to randomly sample the interaction histories for power users or popular items, due to LLM context limits. This is a major information bottleneck. It means the "knowledge" for the most important users and items is being generated from a tiny, random fraction of their data, which could introduce significant noise and instability into the knowledge memory.
- Given the rapid progress in the vision–language modeling (VLM) community, the paper’s experimental setup feels somewhat outdated. Several state-of-the-art and larger-scale VLMs—especially those demonstrating advanced capabilities such as visual reasoning, multi-image understanding, or visual chain-of-thought—are missing from comparison. Incorporating or at least discussing stronger open-source or proprietary baselines (you can find example from the families like OpenAI, Gemini, Claude, Qwen, Hunyuan and so on) would strengthen the evaluation and better position the proposed approach within the current research landscape.

**Questions:**

1. Can the authors provide a practical analysis of the computational cost? How many total LLM calls (and tokens) are required for the "pre-building" phase on the Baby dataset? What is the added latency per batch during training due to the re-ranking (Eq. 13) and knowledge update (Eq. 14) steps?
2. How do the authors envision this framework scaling to a real-world recommender system with millions of users and items? The current approach of running an LLM inference for every user and item seems computationally infeasible.
3. The paper uses a small, fixed $Y$ (e.g., $Y=5$) to sample long histories. How sensitive is the model to this sampling? Is there a risk that the inferred knowledge for power users is unstable or unrepresentative, since it's based on a very small, random sample of their interactions?
4. The "knowledge" ($P_u$, $P_i$) is stored as generated text, this text is then immediately re-encoded by $t_{\theta}(\cdot)$ to be used in graphs (Eq. 5, 10) and GNNs (Eq. 15). Why not have the Integrator Agent output a structured JSON or a set of embeddings directly? What is the benefit of generating natural language text only to immediately embed it again?
5. The feedback loop (Eq. 14) only updates user preferences ($P_u$) and only when a re-ranking fails (produces a worse NDCG). Why not also update item properties ($P_i$)? And why not learn from successful re-rankings as a form of positive reinforcement to strengthen the knowledge?

---

> ### Author Response · Authors · 2025-11-19
>
> Thanks for your comments. We will address your questions point by point.
>
>
>
> **Weakness 1 & Question 1**
>
> > Compared to many previous LLM-based recommendation models, our design of preconstructed homogeneous graphs has already minimized the delays caused by frequent LLM calls during the training phase.
> >
> > In practice, before training, we use the LLM API to extract knowledge for all users and items and refine the traditional item-item graph based on all modalities. This process incurs a total cost of $3(|\mathcal{U}| + |\mathcal{I}|) + |\mathcal{I}|$, which is not significant compared to existing LLM-based recommendation models. Additionally, the graph construction relies solely on similarity calculations, resulting in negligible overhead compared to the overall training costs. It is also worth mentioning that the extracted knowledge can be reused by other models. During the hyperparameter grid search, the knowledge extraction only needs to be performed once beforehand.
> >
> > To further control costs, we opted for a relatively lightweight model, **Qwen2.5-VL-7B**, as the LLM backbone. As shown in **Appendix C.4**, we found that stronger LLM backbones (e.g., **Qwen2.5-VL-32B** and **GPT-4o**) can further enhance the performance of **AgentMMRec**. This indicates that **AgentMMRec** can "evolve" alongside advancements in vision-language models (VLMs).
> >
> > Moreover, to avoid high costs during the training phase, we introduced the hyperparameter **$E$**. In our experiments, we fixed **$E = 10$** (see **Section 4.1** for details).
> >
> > We have provided the number of LLM calls across the three datasets in the revised PDF (highlighted in blue in **Appendix C.6**). The reranking and knowledge update latency depend on the chosen LLM and the hardware setup. For **Qwen2.5-VL-7B**, which is deployed on the same device as AgentMMRec, the latency is very low. In contrast, **Qwen2.5-VL-32B** is deployed on a different device within the same network, resulting in additional latency compared to **Qwen2.5-VL-7B**. For the closed-source model **GPT-4o**, the latency depends on the speed of API access.
> >
> > We have added the end-to-end training time of AgentMMRec under the standard settings on three datasets, compared to the strong baselines, in the appendix (highlighted in blue in **Appendix C.6**).
>
> **Weakness 2**
>
> >  Great question! In fact, AgentMMRec does not heavily rely on the specific templates but instead focuses more on the plan within the template, which provides the detailed task requirements. To address this, we conducted an interesting additional experiment (highlighted in blue in **Appendix C.7**). In this experiment, we regenerated the template five times using three different LLMs and replaced the original templates. We then re-ran the experiments on the all dataset and calculated the mean and variance of the results.
> >
> >  We observed that the mean performance was close to that of the original templates, with very small variance. This indicates that AgentMMRec is not highly sensitive to the specific templates and maintains stable performance as long as the plan design remains consistent.
>
> **Weakness 3 & Question 3**
>
> > We provide experimental results with higher values of $\Upsilon$ in **Figure 4 in Appendix C.3**. **AgentMMRec** benefits from higher $\Upsilon$, but $\Upsilon = 5$ already achieves significant performance advantages. Additionally, our knowledge update process further ensures the quality of the knowledge.

---

> ### Author Response · Authors · 2025-11-19
>
> **Weakness 4**
>
> > We selected the representative open-source model **Qwen2.5-VL-7B** from this year to demonstrate that AgentMMRec remains effective even with smaller-scale large models. In **Appendix C.4**, we provide experimental results using larger open-source LLMs and more powerful closed-source LLMs (e.g., **Qwen2.5-VL-32B** and **GPT-4o**), which further validate that AgentMMRec can benefit from the enhanced capabilities of stronger LLM backbones.
>
> **Question 2**
>
> > In fact, this is entirely feasible. For LLM backbones deployed on the same device or cluster, precomputing millions of LLM calls and similarity matrices is not prohibitively expensive for enterprises. Furthermore, in resource-constrained scenarios, costs can be effectively reduced by performing pre-clustering and using cluster centroids as virtual users/items. All real users/items within a cluster can share the knowledge associated with the cluster centroid.
>
> **Question 4**
>
> > The advantage of our design lies in its ability to be directly integrated into existing multimodal recommendation models (using the same encoder) while ensuring the fairness of experiments (performance improvements come from the model itself, rather than a stronger encoder).
>
> **Question 5**
>
> > When reranking fails, we can evaluate the effectiveness of knowledge updates by examining successful reranking instances. However, after a successful reranking, the effectiveness of the knowledge update cannot be assessed.
>
>
>
> We sincerely appreciate your valuable comments and hope our response has adequately addressed your concerns, providing sufficient reasons to raise the score. We are pleased to address any further concerns you may have. Wishing you all the best!
>
>
> > Friendly reminder: ICLR allows the submission of revised versions during the rebuttal period. Please refer directly to the revised version to review the manuscript updates.

---

> ### Author Response · Authors · 2025-11-26
> **Supplement of response to Reviewer M43Y**
>
> For your convenience, we have provided the updated experimental results in the comments. You can also find all revised parts in the revised version.
>
>
>
> For **End-to-end training time**
>
> > We have added the end-to-end training time of AgentMMRec under the standard settings on three datasets, compared to the strong baselines, in the appendix (highlighted in blue in **Appendix C.6**).
>
> **Table:** Efficiency analysis across all datasets.
>
> | Dataset  | Metrics        | LGMRec | SMORE | MENTOR | COHESION | HPMRec | AgentMMRec |
> | -------- | -------------- | ------ | ----- | ------ | -------- | ------ | ---------- |
> | Baby     | Time (s/epoch) | 5.93   | 6.55  | 7.03   | 4.47     | 21.03  | 6.04       |
> |          | Memory (GB)    | 2.41   | 3.31  | 7.12   | 2.89     | 8.58   | 3.07       |
> | Sports   | Time (s/epoch) | 8.98   | 9.29  | 9.62   | 7.91     | 30.85  | 9.13       |
> |          | Memory (GB)    | 3.67   | 5.02  | 8.44   | 4.20     | 10.19  | 4.48       |
> | Clothing | Time (s/epoch) | 10.02  | 11.05 | 11.90  | 9.05     | 40.23  | 10.65      |
> |          | Memory (GB)    | 4.81   | 6.89  | 12.99  | 5.73     | 14.95  | 5.72       |
>
>
>
> For **Template dependencies**
>
> > AgentMMRec does not heavily rely on the specific templates but instead focuses more on the plan within the template, which provides the detailed task requirements. To address this, we conducted an interesting additional experiment (highlighted in blue in **Appendix C.7**). In this experiment, we regenerated the template five times using three different LLMs and replaced the original templates. We then re-ran the experiments on the all dataset and calculated the mean and variance of the results.
> >
> > We observed that the mean performance was close to that of the original templates, with very small variance. This indicates that AgentMMRec is not highly sensitive to the specific templates and maintains stable performance as long as the plan design remains consistent.
>
> **Table:** Performance comparison of AgentMMRec using different templates regenerated five times by various LLMs, evaluated in terms of Recall@20.
>
> | Datasets          | Baby   |        | Sports |        | Clothing |        |
> | ----------------- | ------ | ------ | ------ | ------ | -------- | ------ |
> | Metrics           | Mean   | Var    | Mean   | Var    | Mean     | Var    |
> | Origin            | 0.1079 | -      | 0.1231 | -      | 0.1071   | -      |
> | GPT-5             | 0.1082 | 0.0012 | 0.1228 | 0.0015 | 0.1070   | 0.0010 |
> | Claude-Sonnet-4.5 | 0.1077 | 0.0008 | 0.1230 | 0.0008 | 0.1074   | 0.0012 |
> | Gemini-2.5-Pro    | 0.1080 | 0.0013 | 0.1228 | 0.0011 | 0.1068   | 0.0006 |
>
> For **LLM costs**
>
> > We further discuss the cost of using LLM during the pre-training graph construction phase. When extracting knowledge for users and items, we make $3(|\mathcal{U}| + |\mathcal{I}|)$ calls to the LLM. Additionally, for the refinement of the item-item graph, we require another $|\mathcal{I}|$ calls.
>
> **Table:** LLM costs for all datasets.
>
> | Datasets        | Baby   | Sports  | Clothing |
> | --------------- | ------ | ------- | -------- |
> | Building Graphs | 79,485 | 161,865 | 187,260  |
> | Refining Graphs | 7,050  | 18,357  | 23,033   |
> | Total           | 86,535 | 180,222 | 210,293  |
>
>
> This is a supplement to the above rebuttal. All updates can be found in the revised version.
>
> We sincerely appreciate your valuable comments and hope our response has adequately addressed your concerns, providing sufficient reasons to raise the score. We are pleased to address any further concerns you may have, and we wish you all the best!

---

### Official Review · Reviewer_AxGP · 2025-11-01

**Soundness:** 2
**Presentation:** 2
**Contribution:** 2
**Rating:** 4
**Confidence:** 4

**Summary:**

AgentMMRec introduces an original and ambitious concept—treating LLMs as cooperating knowledge agents for multimodal recommendation. The work shows solid empirical results and good system design, representing a step toward integrating reasoning and recommendation.

**Strengths:**

1. The paper is well-written, includes extensive analyses (cold-start, hyperparameter, backbone LLM, multimodal alignment), and provides open-source code.
2. The semantic gap and noise issues between modalities are critical challenges in multimodal operating systems.

**Weaknesses:**

1. The overall methodological novelty is limited. Although the system is well integrated, its core ideas mainly extend existing LLM-based recommendation frameworks rather than introducing fundamentally new mechanisms.
2. The model’s performance heavily depends on the feedback and reasoning quality of the underlying LLM, raising concerns about robustness and reproducibility across different backbones.
3. The approach incurs substantial computational and resource costs, while yielding only minor performance gains.
4. Experiments are conducted on small-scale datasets, leaving scalability and stability on large, real-world multimodal recommendation scenarios unverified.
5. The comparison with existing methods is incomplete. Several recent studies (particularly in 2025) have highlighted the significant impact of noise in multimodal recommendation, yet these works were not included in the comparison, making the experimental evaluation less comprehensive and representative.

**Questions:**

See the issues discussed in the “Weaknesses” section above.



Please further clarify the core novelty of the proposed method. Specifically:
1. What is the distinct innovation of this framework compared to existing multimodal recommendation or knowledge-enhanced graph models?
2. Why is the use of large language models (LLMs) indispensable? Could a lighter-weight semantic reasoning or knowledge extraction module achieve similar effects?
3. If LLMs mainly serve as information integrators or alignment modules, their necessity and advantage should be empirically justified, not only conceptually stated.

---

> ### Author Response · Authors · 2025-11-19
>
> Thanks for your comments. We will address your questions point by point.
>
> **Weakness 1 & Question 1**
>
> > We would like to clarify the novelty of AgentMMRec. Compared to existing LLM-based recommendation frameworks, AgentMMRec introduces the following unique innovations:
> >
> > 1. This novel design ensures that LLMs can fully utilize their multimodal understanding and contextual reasoning capabilities to better align multimodal information with recommendation tasks. Additionally, the preconstructed graphs based on raw features and historical interactions require minimal updates during training, significantly reducing the inherent high computational overhead of LLM-based recommender systems. Our AgentMMRec is both efficient and unique, distinguishing it from prior works.
> > 2. Unlike previous works, our reranking mechanism is not only used to reorder the final recommendation list but also to enhance the quality of the maintained knowledge. While constructing homogeneous graphs, we simultaneously maintain a corresponding knowledge memory, which is utilized during the reranking stage. The evaluation from reranking is further used to refine and optimize the knowledge quality, forming a cyclic evolutionary process. This innovative design, integrating reranking with knowledge optimization, is a key feature absent in prior works.
> > 3. Unlike existing LLM-based recommendation models, the homogeneous graphs we construct can be directly integrated into existing multimodal recommendation models, resulting in significant performance improvements (see **Table 2 in Section 4.4**). Furthermore, the maintained knowledge memory can be seamlessly transferred to existing models and even jointly optimized across multiple models (see **Table 5 in Appendix C.2**). These designs and advantages are unique to our approach and not present in previous works.
>
> **Weakness 2**
>
> > We provide an analysis of different LLM backbones in **Table 6 in Appendix C.4**. Notably, in all other experiments, we use a relatively lightweight LLM model, **Qwen2.5-VL-7B** (as detailed in **Section 4.1**). Therefore, AgentMMRec can further benefit from leveraging more powerful LLM backbones.
>
> **Weakness 3**
>
> > I suspect that your inference is based solely on the results from the baby dataset. I hope the results on the sports and clothing datasets will also capture your attention. In fact, the performance improvement of AgentMMRec is significant, breaking the performance barriers of recent multimodal recommendation systems by fundamentally improving the quality of data and graph structures. Please refer to **Table 1** and the discussion in **Section 4.2** (Observation 2). Notably, we selected some of the highest-performing baselines among open-sourced multimodal recommendation systems available as of the ICLR submission deadline. For more information, please refer to the latest survey [1] and its continuously updated awesome list repository.
> >
> > Moreover, as we mentioned in our response to **Weakness 1**, AgentMMRec does not incur significant additional costs. Unlike existing LLM-based recommendation systems, the graph construction in AgentMMRec is performed before the training stage, based on raw features and historical interactions, which avoids the high computational costs typically associated with training. Additionally, for efficiency, we set the knowledge update interval to E = 10, as detailed in **Section 4.1**.
>
> **Weakness 4**
>
> > We selected the three most commonly used and representative datasets in multimodal recommendation, which are utilized by nearly all state-of-the-art multimodal works. This choice ensures fairness in comparison.

---

> ### Author Response · Authors · 2025-11-19
>
> **Weakness 5**
>
> > We are aware of some works focusing on robustness; however, these are implemented as components rather than standalone models and were therefore not included in our comparison (e.g., [2]).
> >
> > For our baselines, we selected some of the highest-performing open-sourced multimodal recommendation systems available up to the ICLR submission deadline.
> >
> > If you have additional suggestions for open-source baselines, please let us know, and we will incorporate them into our evaluation as soon as possible.
>
> **Question 2**
>
> > This is due to the unique advantages of LLMs in context processing and multimodal understanding. We believe these two strengths are highly aligned with the requirements of multimodal recommendation tasks. The context processing capability allows LLMs to understand the relationships between users, items, and their historical interactions, while the multimodal understanding ability enables effective alignment of multimodal information. These two aspects have long been significant challenges in recommendation tasks, particularly in multimodal recommendation.
> >
> > Furthermore, the combination of these two strengths in LLMs can effectively bridge the gap between multimodal information and recommendation tasks. In contrast, lightweight semantic reasoning or knowledge extraction modules struggle to integrate historical interaction data with multimodal understanding effectively, leaving the inherent disparity between multimodal information and recommendation tasks unresolved.
>
> **Question 3**
>
> > We leverage both the context processing and multimodal understanding capabilities of LLMs, as well as the effectiveness of the knowledge-based graph structure and reranking process. The effectiveness of these components has been empirically validated through ablation studies (see **Section 4.3**).
> >
> > Additionally, we provide a compatibility analysis of the constructed homogeneous graph and reranking process when transferred to other models (see **Section 4.4**). To further support our findings, we also conducted interesting experiments on jointly training the knowledge memory with multiple models (see **Appendix C.2**).
>
> Refs
>
> [1] A Survey on Multimodal Recommender Systems: Recent Advances and Future Directions, TMM 2025.
>
> [2] Enhancing Robustness and Generalization Capability for Multimodal Recommender Systems via Sharpness-Aware Minimization, TKDE 2025.
>
> We sincerely appreciate your valuable comments and hope our response has adequately addressed your concerns, providing sufficient reasons to raise the score. We are pleased to address any further concerns you may have. Wishing you all the best!
>
>
>
> > Friendly reminder: ICLR allows the submission of revised versions during the rebuttal period. Please refer directly to the revised version to review the manuscript updates.

---

> ### Author Response · Authors · 2025-11-27
> **Supplement of response to Reviewer AxGP**
>
> For **Weakness 5**
>
> > We revisited the papers on denoising for multimodal recommendation published in 2025 and identified two works: FreRec [1] and EVEN [2]. Unfortunately, neither of these works has been open-sourced. To address your concerns, we have directly reported the experimental results from the papers.
>
> > Please refer to Table 1 highlighted in blue in the revised version.
>
> > We hope that the addition of these experimental results can alleviate your concerns regarding the comprehensiveness and representativeness of the evaluation.
>
> We sincerely appreciate your valuable comments and hope our response has adequately addressed your concerns, providing sufficient reasons to raise the score. We are pleased to address any further concerns you may have. Wishing you all the best!
>
> Refs:
>
> [1] Frequency-refined Graph Convolution Network with Cross-modal Wavelet Denoising for Recommendation, ACM MM 2025.
>
> [2] Seeing Beyond Noise: Joint Graph Structure Evaluation and Denoising for Multimodal Recommendation, AAAI 2025.

---

### Comment · Area_Chair_7x5H · 2025-11-25

Dear reviewers,
Thank you for your contribution to the ICLR26 review. After reading the authors' rebuttal and other reviewers' comments, please can you engage in the discussion (by replying to this thread) if you change your mind or not?
Thanks, Your AC

---

### Author Response · Authors · 2025-11-27

Dear Reviewers,

We sincerely appreciate your valuable comments.

We hope that you find our detailed responses helpful. We have thoroughly addressed each of your comments and would be happy to clarify anything further if needed. As the discussion period will end in less than a week, we would be very grateful if you could kindly take a moment to review our responses.

Looking forward to your positive feedback and to discussing the work further if needed.

Best regards,
Authors of ICLR 13942

---

### Author Response · Authors · 2025-11-29
**Summary of revisions**

Dear AC, SAC, and PC,

We sincerely thank all reviewers for their insightful and constructive comments, despite some unexpected circumstances that arose during the rebuttal period.

We fully recognize the significant responsibilities and workload that the AC has managed under these conditions. To facilitate the review process, we have prepared a concise summary of the revisions made during this time.

In addition, we have provided comprehensive responses to all reviewer concerns and conducted several new experiments to support our claims.

Although further discussion with the reviewers was not possible due to the situation, we believe our updated responses and additional results adequately address all the concerns raised.

In summary, we have incorporated the following key additions and revisions:

- Two new baselines (Section 4.1 and Table 1)
- Efficiency study (Appendix C.6)
- Template dependency analysis (Appendix C.7)

To ensure that the AC, SAC, PC, and all reviewers can easily verify the revisions, all changes have been thoroughly documented in the revised version and are clearly marked in blue.

Thank you once again for your time and consideration.

Warm regards,

Authors of ICLR 13942

---

### Meta-Review · Area_Chair_kW66 · 2026-01-11

**Summary:**

This paper proposes AgentMMRec, an LLM-based agent framework with two cooperative agents (Integrator and Utilizer) to bridge the semantic gap in multimodal recommendation. Experiments on three datasets demonstrate its superiority over existing models, especially in data sparsity scenarios.

The reviewers' key concerns lie in: limited methodological novelty, high computational cost, poor scalability, and heavy reliance on prompt templates.

**Reviewer Concerns:**

Addressed Concerns:
- Computational cost: Added efficiency study (training time, memory usage, LLM call counts).

- Prompt template reliance: Conducted template dependency analysis with multiple LLMs, proving stable performance.

- Incomplete experiments: Supplemented results of 2025 denoising-related baselines and compared with stronger VLMs (Qwen2.5-VL-32B, GPT-4o).

Outstanding Concerns:
- Methodological novelty: partly addressed with clarified innovations (preconstructed graphs, reranking-knowledge update loop, model-agnostic knowledge transfer).

- Scalability to large-scale real-world scenarios (millions of users/items) remains unproven.

- Limited modality coverage (only text+image) is unaddressed.

**Reviewer Scores:**

- Reviewer AxGP: Original score 4 (marginally below acceptance), likely to maintain the score due to the novelty issues.

- Reviewer M43Y: Original score 6 (marginally above acceptance), likely to remain stable due to the comprehensive response to computational cost and template dependency concerns.

- Reviewer HDYt: Original score 4 (marginally below acceptance), likely to maintain the score due to the novelty issues.

- Reviewer 4vcD: Original score 4 (marginally below acceptance), likely to improve the score due to the added training time comparison and data leakage clarification.

---

### Decision · Program_Chairs · 2026-01-26

Reject